# *Duolando*: Follower GPT with Off-Policy Reinforcement Learning for Dance Accompaniment

**Li Siyao**[1]    **Tianpei Gu**[2*]    **Zhengyu Lin**[3]    **Zhitao Yang**[3]    **Ziwei Liu**[1]
**Henghui Ding**[1]    **Lei Yang**[3,4 ✉]    **Chen Change Loy**[1 ✉]
[1]S-Lab, Nanyang Technological University   [2]Lexica   [3]SenseTime   [4]Shanghai AI Laboratory
https://lisiyao21.github.io/projects/Duolando

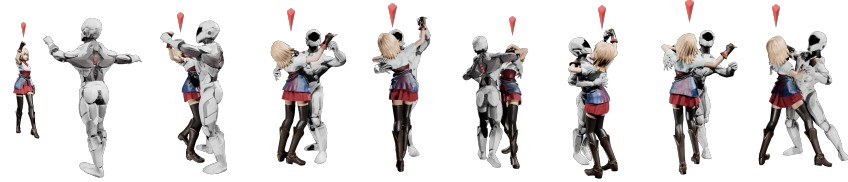

Figure 1: **Example of *Duolando*'s results.** The female avatar (red arrow) is driven by the proposed method to accompany real human's (white) dancing.

## Abstract

We introduce a novel task within the field of 3D dance generation, termed *dance accompaniment*, which necessitates the generation of responsive movements from a dance partner, the "follower", synchronized with the lead dancer's movements and the underlying musical rhythm. Unlike existing solo or group dance generation tasks, a duet dance scenario entails a heightened degree of interaction between the two participants, requiring delicate coordination in both pose and position. To support this task, we first build a large-scale and diverse duet interactive dance dataset, *DD100*, by recording about 117 minutes of professional dancers' performances. To address the challenges inherent in this task, we propose a GPT-based model, ***Duolando***, which autoregressively predicts the subsequent tokenized motion conditioned on the coordinated information of the music, the leader's and the follower's movements. To further enhance the GPT's capabilities of generating stable results on unseen conditions (music and leader motions), we devise an off-policy reinforcement learning strategy that allows the model to explore viable trajectories from *out-of-distribution* samplings, guided by human-defined rewards. Based on the collected dataset and proposed method, we establish a benchmark with several carefully designed metrics.

## 1 Introduction

Duet dancing is an interactive art involving coordination between two individuals' body movements under background music. In the context of ballroom dancing, the duet dancers typically comprise a *leader* and a *follower*: the leader is responsible for initiating and guiding the movements, while the follower responds to the leader's cues and follows his/her lead. Developing a computational model that enables virtual agents to accompany the dance with the user-controlled leader has great potential in a wide range of virtual reality (VR) and augmented reality (AR) applications.

Although the dance accompaniment task has broad academic and practical value, there is no currently available duet dance data in public. To facilitate this task, we build a large-scale dataset named DD100 (***DuetDance100***). Specifically, we record data of 10 different genres of ballroom dancing, including 5 kinds of Latin (Cha Cha, Rumba, Samba, Paso Doble and Jive), 4 Moderns (Foxtrot, Tango, Quickstep and Waltz) and *Pas de Deux* of ballet performed by 5 pairs of professional dancers under unique background music. For each genre, the performers play 10 distinct clips, within a duration of 36 to 107 seconds, resulting in a total duration of around 117 minutes. As duet dances usually involve significant parts of occlusion and rotations, we collect the 3D motion data using professional MoCap equipment to ensure data quality. The final data consists of SMPL-X (Pavlakos et al., 2019) sequences, with body poses reconstructed from the point clouds by and hand gestures

---

✉ Corresponding authors. * Work mostly completed at UCLA.

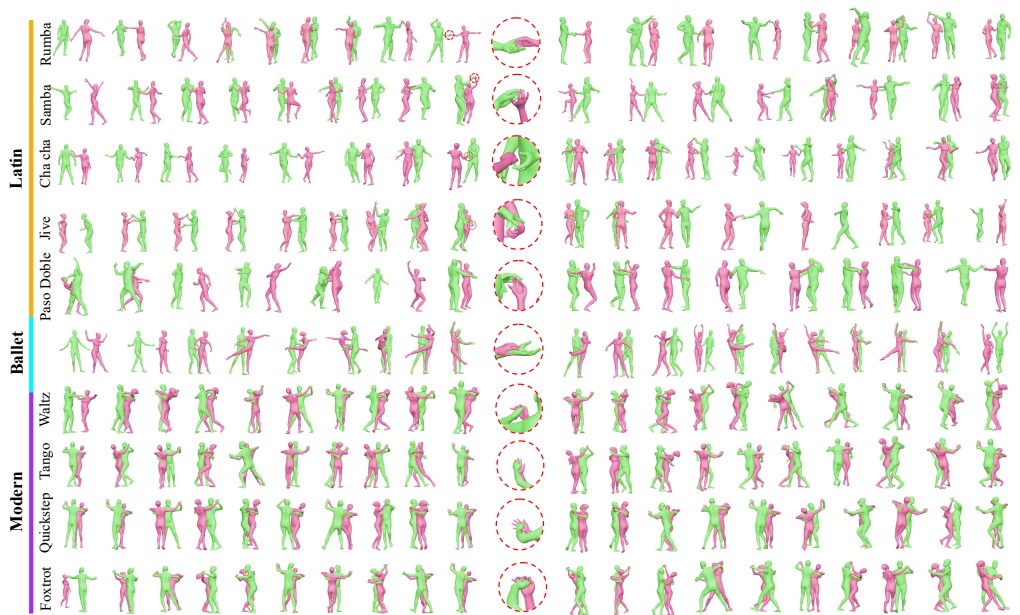

Figure 2: **Samples of `DD100` dataset.** The leader and the follower are colored in green and red, respectively. `DD100` contains 10 dance genres, featuring a diverse range of poses and interactions, with intricate hand gestures.

deduced from meta glove records. `DD100` provides a rich context of interactive coordination between two dancers, including either physical connections (such as steps with hand holding) or strong semantic relations (such as leader-centered surrounding), which serves as a fundamental basis for the study of related to duet interactive dance.

Despite having a high-quality duet dance dataset, the dance accompaniment task cannot be solved by applying existing solo dance methods (Ren et al., 2020; Sun et al., 2020; Vaswani et al., 2017; Li et al., 2021b; Siyao et al., 2022; Sun et al., 2022; Tseng et al., 2023). As an interactive art, a duet dance not only requires the follower to maintain the aesthetics and the sense of rhythm of his/her own movements, but also demands a high level of interactive coordination with the leader. Such requirements make existing works on solo dancing inadequate for this task as they lack the scope of the partner. Meanwhile, generating a single agent's motion in response to a pre-conditioned leader's movement presents more significant challenges than simultaneously synthesizing multiple agents (Le et al., 2023), since the latter can retrieve well-coordinated motion patterns from the training data, which simplifies the task.

In this paper, we propose a two-stage framework to generate follower motion in harmony with both the background music and the leader's movements. In the first stage, we train VQ-VAEs to embed and quantize the dance movements of different body parts and the relative translation between the two dancers. Then, in the second stage, we devise an interaction-coordinated GPT to autoregressively predict the next token, conditioned on the amalgamated information of the music signal, the leader's motion, and the previous follower sequence. Stability issues arise when confronted with unheard music or unseen leader motion patterns. A common observation is that the lower body movement appears incompatible with the global displacement, resulting in skating artifacts. To address this challenge, we introduce an off-policy reinforcement learning strategy for GPT, enabling our method to handle out-of-distribution instances robustly based on human-defined rewards. The above modules contribute to *Duolando*, which can generate reasonable movement responding to the leader's movement and serves as a strong baseline of dance accompaniment.

The contributions of our paper are three-fold: **(1)** We introduce a novel multi-modal task, dance accompaniment, and provide a large-scale and diverse dataset for both training and testing purposes. Leveraging this data, we establish a new benchmark with metrics reflecting both the quality of isolated dancers and the interaction between partners. **(2)** We construct a GPT-based network capable of generating motion sequences, taking into account the coordination between partners, which serves as a robust baseline for this task. **(3)** We introduce an off-policy reinforcement learning strategy for GPT to address out-of-distribution challenges, and demonstrate its successful application in our task.

Table 1: **Comparison with human-human interaction and music-to-dance datasets.** HHI denotes Human-Human Interaction, where S represents *Strong* interaction with physical contact while W means *Weak* interaction like repeated motion in group dance. ♫ indicates whether having accompanied music modality. ✋ denotes whether having hand (finger-level) motions. # Subj. denotes the number of performers. $\overline{T}$ denotes average duration and $T$ is the total duration of all sequences. MV stands for capturing with multi-view cameras. Genres for human-human interaction dataset means the type of interactions, while it indicates the music and dance styles for music-to-dance datasts. n/a means that the exact information is missed in the original paper. *Interactions without physical contact appear in partial data (Showcase, Cypher and Battle) in AIST++.

| Dataset | HHI | ♫ | ✋ | # Genres | # Subj. | $\overline{T}$ | $T$ | Acquisition | GT |
|---|---|---|---|---|---|---|---|---|---|
| CMU-MoCap (interactive) (cmu) | W | ✗ | ✗ | 10 | 8 | 5.2s | 285.5s | MoCap | 3D Joints |
| UMPM (Van der Aa et al., 2011) | W | ✗ | ✗ | 7 | 30 | 222s | 2.2h | MoCap | 3D Joints |
| NTU-RGB+D 120 (Liu et al., 2020) | S | ✗ | ✗ | 26 | 106 | 2.7s | 0.47h | Kinect | 3D Joints |
| You2Me (Ng et al., 2020) | S | ✗ | ✗ | 4 | 10 | 120s | 1.4h | MV & Kinect | 3D Joints |
| InterHuman (Liang et al., 2023) | S | ✗ | ✗ | 11 | n/a | 3.9s | 6.56h | MV | SMPL |
| DanceNet (Zhuang et al., 2022) | ✗ | ✓ | ✓ | 2 | 2 | n/a | 0.96h | MoCap | 3D Joints |
| Dancing2Music (Lee et al., 2019) | ✗ | ✓ | ✗ | 3 | n/a | 6s | 71h | Pseudo | 2D Joints |
| DanceRevolusion (Huang et al., 2021) | ✗ | ✓ | ✗ | 3 | n/a | 60s | 12h | Pseudo | 2D Joints |
| PMSD (Valle-Pérez et al., 2021) | ✗ | ✓ | ✓ | 3 | 1 | n/a | 3.1h | MoCap | 3D Joints |
| SMVR (Valle-Pérez et al., 2021) | ✗ | ✓ | ✗ | 2 | 8 | n/a | 9h | VR Tracker | 3D Joints |
| AIST++ (Li et al., 2021b) | W* | ✓ | ✗ | 10 | 30 | 13s | 5.2h | Pseudo | SMPL |
| AIOZ-GDANCE (Le et al., 2023) | W | ✓ | ✗ | n/a | 4000+ | 37.5s | 16.7h | Pseudo | SMPL |
| DD100 | S | ✓ | ✓ | 10 | 10 | 70.2s | 1.95h | MoCap | SMPL-X |

## 2 DD100: A LARGE-SCALE DUET DANCE MOCAP DATASET

**Data Statistics.** To facilitate research in the dance accompaniment task, a large-scale dataset of duet dances, named DD100, was collected. DD100 comprises ten distinct genres of duet dances, all featuring strong interaction between the dancers. The genres include five Latin dances (Cha Cha, Rumba, Samba, Paso Doble, and Jive), four Modern (a.k.a. Standard) dances (Foxtrot, Tango, Quickstep, and Waltz), and *Pas-de-Deux* ballet. The dances were performed by five pairs of professional dancers, while each dance sequence was accompanied by unique background music. Each dance genre was recorded by ten distinct clips, with clip durations ranging from 49 to 96 seconds, resulting in a total duration of approximately 1.9 hours (or approximately 115.4 minutes). In the experiment, we randomly split the dataset into the 80% training set and 20% test set, with the training set of 168,176 frames (5605.9 seconds) and the test set of 42,496 frames (1416.5 seconds). During the recording, nearly 80% sequences are performed twice. If counting these duplicated takes, the total duration will be 3.24 hours. Considering the diversity in moving direction and difference in motion details at each take, we use all these data for training and testing in practice.

**Data Collection.** To obtain high-quality data, we used 20 optical MoCap cameras to capture body data for two dancers at 120 FPS. The raw Mocap data consist of the 3D positions of 53 marker points on body surface in each frame. To process these data into SMPL-X (Pavlakos et al., 2019) format, we first select a clip for each dancer to fit his/her body shape parameters (`beta`), following the pipeline of SOMA (Ghorbani & Black, 2021). Then, we use the pre-processed body shape to assist pose parameters (`theta`) regression in each frame after filtering out those invisible points with confidence scores lower than a threshold to keep the robustness of the regression results. During this procedure, we do not fit the hand movement. Since the hands are prone to self-occlusion or inter-occlusions between two individuals, we employed inertial motion capture gloves (meta gloves) to capture these data. The raw data of meta gloves are stored in BioVision Motion Capture (BVH) format, describing hierarchical rotations of finger joints in Euler angles. We transfer those Euler angles to axis angles aligning the MANO (Romero et al., 2022) model initialized with "flat" gestures for both hands. As the "flat" pose of MANO differs from the initial one in BVH format with fingers apart in specific angles, we subtract this difference while mapping from BVH to MANO. Finally, we combine the body and hand parameters with the wrist rotations from MoCap regression. Each dance clip data in DD100 consists of the SMPL-X (Pavlakos et al., 2019) sequences for both the leader and follower, along with the corresponding music within an average rhythmical beat of 118 beat per minute (BPM) from 72 BPM to 163 BPM.

**Comparison to Related Datasets.** As shown in Table 1, DD100 stands out from existing datasets by simultaneously incorporating the following distinct features: **(1) Complex and Strong Interaction.** Existing solo dance datasets (Zhuang et al., 2022; Lee et al., 2019; Huang et al., 2021; Valle-Pérez et al., 2021; Li et al., 2022) overlook the element of interaction, while existing group dance

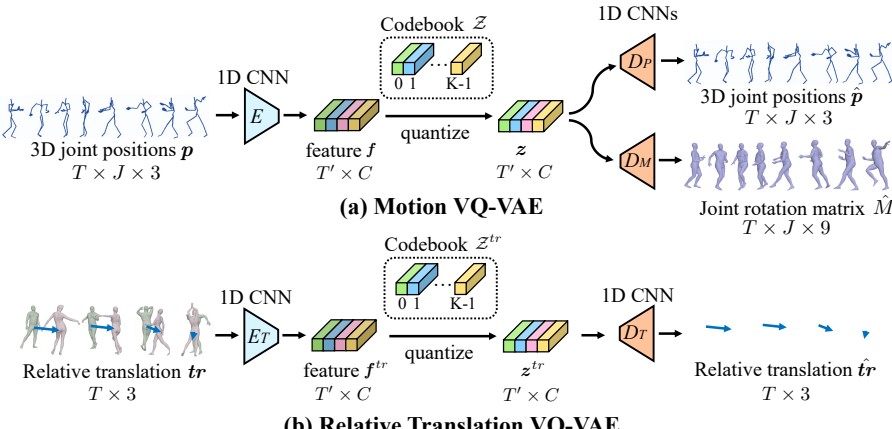

Figure 3: **(a) Structures of Motion VQ-VAEs and (b) Relative Translation VQ-VAE.** The quantization is to substitute a encoded feature to the most similar one $z_k$ in the codebook $\mathcal{Z}$ such that $z_k = \arg\min_{z \in \mathcal{Z}} \|f_i - z\|$.

datasets (Le et al., 2023) primarily emphasize group movement coordination with similar motions replicated to different agents with minimal physical contact. Furthermore, current human-human interaction datasets (cmu; Van der Aa et al., 2011; Liu et al., 2020) typically feature simpler actions and weaker interactions, such as handshakes and hugs. In contrary, as depicted in Figure 2, duet dances in `DD100` involve more complex motions and intense interactions between various body parts. **(2) Multimodal.** Unlike datasets (Ng et al., 2020; Liang et al., 2023) focusing on human-human interaction, `DD100` incorporates a musical modality, which is a crucial factor to consider during the generation process to ensure synchrony with the music. **(3) Extended Duration.** Many existing human motion datasets feature short durations. For example, dances in AIST++ (Li et al., 2021b) typically last around 10 seconds. In contrast, each dance clip in `DD100` extends over one minute. This not only supplies a higher-level choreography reference for model training but also poses the proposed dance accompaniment problem to be more realistic and challenging.

## 3 OUR APPROACH

To address the dance accompaniment task, we propose a baseline method, *Duolando*, a GPT-based network enhanced with off-policy reinforcement learning to improve its generalization ability.

### 3.1 QUANTIZING MOTION AND RELATIVE TRANSLATION

Previous studies (Siyao et al., 2022; Ng et al., 2022) has viewed dance as a sequential combination of reusable dance positions, capturing the essence of movement in a rhythmic context. To efficiently distill and quantize these basic dance components into a deep feature space in an unsupervised manner, we draw on VQ-VAE (Van Den Oord et al., 2017). This provides a tokenized and interpretable representation of the dance motion sequence. Our approach incorporates a 1D CNN-based VQ-VAE to preserve the spatio-temporal continuity of dance movement. Inspired by Siyao et al. (2022), we use four VQ-VAEs to encode and quantize the motion of four body parts (upper half body, lower half body, left hand, and right hand) into code sequences $z^{up}$, $z^{down}$, $z^{lhand}$, and $z^{rhand}$. Additionally, we employ another VQ-VAE $VQ^{tr}$ to model the relative translation $tr$ between the follower and leader. This allows the GPT model to explicitly output the relative positions of the follower by subtracting the root point of the follower from that of the leader. Please see Figure 3 for details.

In the case of motion VQ-VAEs, a $T \times J \times 3$ sequence of 3D joint positions is fed into a 1D-CNN encoder $E$, where $T$ is the frame length and $J$ is the joint count. This is then translated to a temporally downsampled deep feature $f \in \mathbb{R}^{T' \times C}$, where $T' = T/d$ and $C$ is the number of channels. Subsequently, deep feature $f$ is quantized into code sequence $z$ by replacing each element $f_i$ with an element $z_k$ in codebook $\mathcal{Z}$ that shows the smallest difference $\|f_i - z_k\|$. For training the motion VQ-VAE, we use two decoders, $D_P$ and $D_M$, which translate the quantized code back to 3D joint positions $\hat{p}$ and rotation matrix $\hat{M}$, respectively, allowing us to generate rotations that can directly animate avatars without the need for inverse kinematics. The training loss of the motion

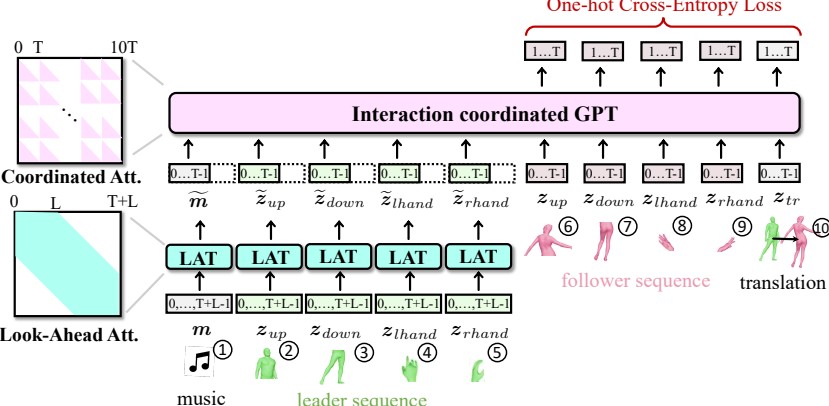

Figure 4: **Structure of follower GPT.** The GPT takes ten inputs and autoregressively predicts the subsequent tokens of follower's motion and the relative translation. Preconditions (music signals and leader's motion) are integrated with Look-Ahead Transformers (LAT).

VQ-VAE is computed as follows:

$$\mathcal{L}_{VQ} = \mathcal{L}_{rec}(\hat{\boldsymbol{p}}, \boldsymbol{p}) + \mathcal{L}_{rec}(\hat{M}, M) + \|\text{sg}(\boldsymbol{f}) - \boldsymbol{z}\| + \lambda\|\boldsymbol{f} - \text{sg}(\boldsymbol{z})\|, \quad (1)$$

where $\mathcal{L}_{rec}$ is the sum of $l_1$-reconstruction losses for the values, the first (velocity) and the second derivatives (acceleration) of the reconstructed motion sequence and the ground truth. $\boldsymbol{p}$ and $M$ denote the ground-truth 3D joint positions and rotation matrix, respectively, while "sg" denotes "stop gradient" (Chen & He, 2021) and $\lambda$ is the trade-off "commitment" parameter (Dhariwal et al., 2020; Siyao et al., 2022). The training loss of the relative translation VQ-VAE follows the same formula but only includes one reconstruction loss, specifically $\mathcal{L}_{rec}(\boldsymbol{tr}, \hat{\boldsymbol{tr}})$.

### 3.2 Interaction Coordinate GPT

**Formulation.** With the pretrained VQ-VAE, the 3D dancing movement can be transferred into a sequence of quantized code numbers. The goal of using GPT is to generate the follower's dance sequence in the quantized code domain with the maximum likelihood:

$$(\boldsymbol{z}^{up\heartsuit}, \boldsymbol{z}^{down\heartsuit}, \boldsymbol{z}^{lhand\heartsuit}, \boldsymbol{z}^{rhand\heartsuit}, \boldsymbol{z}^{tr}) = \arg\max_{\boldsymbol{z}} \Pr(\boldsymbol{z}|\boldsymbol{m}, \boldsymbol{z}^{up\spadesuit}, \boldsymbol{z}^{down\spadesuit}, \boldsymbol{z}^{lhand\spadesuit}, \boldsymbol{z}^{rhand\spadesuit}). \quad (2)$$

We use shared VQ-VAEs to generate the code sequences for the dancers. The leader's movement is represented in green ($\boldsymbol{z}^{\spadesuit}$) with notion $\spadesuit$ while the follower's movement is represented in pink ($\boldsymbol{z}^{\heartsuit}$) with notion $\heartsuit$. The generated code sequences are then decoded by $D_M(\boldsymbol{z}^{\heartsuit})$ to reconstruct the 3D joint rotation and drive the follower's motion with global root position $\boldsymbol{q}^{\heartsuit}$ computed by $\boldsymbol{q}^{\heartsuit} = D_T(\boldsymbol{z}^{tr}) + \boldsymbol{q}^{\spadesuit}$, where $\boldsymbol{q}^{\spadesuit}$ is the root position of the leading dancer. For more details on the follower GPT structure, refer to Figure 4.

**Looking-Ahead Conditions.** In a duet dance, the follower needs not only to respond to the leader's current movements but also to anticipate future changes in the dance dynamics. To fulfill this requirement, we implement a *look-ahead* mechanism that allows the GPT to be aware of future conditional signals. Specifically, when sampling the conditional inputs (music $\boldsymbol{m}$ and leader motion $\boldsymbol{z}^{\spadesuit}$), we extract an additional $L$ tokens beyond the GPT's block size $T$, leading to conditioning sequences of length $(T + L)$. These extended conditioning sequences are then processed through Look-Ahead Transformers (LAT) to derive deep embeddings, denoted as $\widetilde{\boldsymbol{m}} = \text{LAT}(m_{0,...,T+L-1})$ and $\widetilde{\boldsymbol{z}}^{\spadesuit} = \text{LAT}(z^{\spadesuit}_{0,...,T+L-1})$. Within the LATs, we use look-ahead attention layers to propagate information from $L$ future tokens to the current one, which is achieved through a banded mask of attention with a window size of $L$. By incorporating future rhythms and leader movements, the GPT predicts more synchronized and stable follower dance positions.

**Interaction Coordination.** After the look-ahead expansion is completed, we truncate the first $T$ embeddings of $\widetilde{\boldsymbol{m}}$ and $\widetilde{\boldsymbol{z}}^{\spadesuit}$, and use them together with the follower motion $\widetilde{\boldsymbol{z}}^{\heartsuit}$ and relative translation $\boldsymbol{z}^{tr}$ indexed from 0 to $T-1$ as input for our model. The GPT then incrementally generates predictions

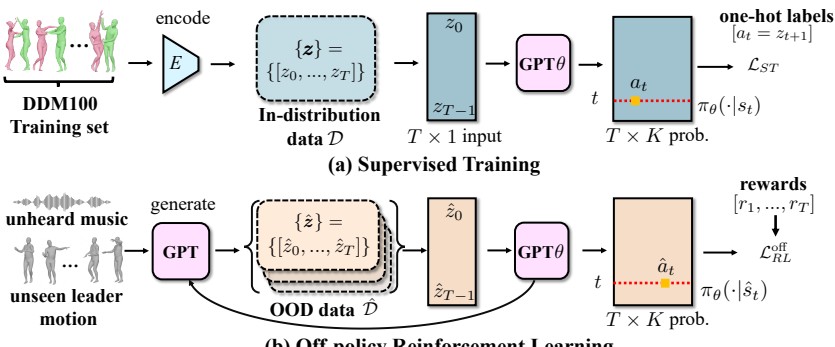

Figure 5: **GPT's supervised training (a) *vs* off-policy RL stage (b).** For ST, supervising labels are sequences $\{z\}$ quantized from the in-domain data in training set. In RL stage, network weight is optimized based on human-defined rewards (scores) on GPT's generated samples $\{\hat{z}\}$ on OOD conditions.

for subsequent tokens, which correspond to indices ranging from $1$ to $T$. Formally,

$$
\begin{aligned}
& z^{up\heartsuit}_{1,...,T}, z^{down\heartsuit}_{1,...,T}, z^{lhand\heartsuit}_{1,...,T}, z^{rhand\heartsuit}_{1,...,T}, z^{tr}_{1,...,T} \\
= \ & \mathrm{GPT}(\widetilde{m}_{0,...,T-1}, \widetilde{z}^{up\spadesuit}_{0,...,T-1}, \widetilde{z}^{down\spadesuit}_{0,...,T-1}, \widetilde{z}^{lhand\spadesuit}_{0,...,T-1}, \widetilde{z}^{rhand\spadesuit}_{0,...,T-1}, \\
& z^{up\heartsuit}_{0,...,T-1}, z^{down\heartsuit}_{0,...,T-1}, z^{lhand\heartsuit}_{0,...,T-1}, z^{rhand\heartsuit}_{0,...,T-1}, z^{tr}_{0,...,T-1}).
\end{aligned}
\tag{3}
$$

To effectively integrate the information of the 10 input items in Equation 3, we use a 10-by-10 block-wise lower-triangular matrix as the mask of interaction-coordinated attention. Since the lower triangular matrix masks off the future information, this preserves the causal order of inference, ensuring that past tokens do not have access to future ones, and maintain the integrity and coherence of these items during the prediction process.

### 3.3 OFF-POLICY GPT REINFORCEMENT LEARNING

Although GPT is powerful, it is still prone to generate unsatisfactory results when confronted with *out-of-distribution* (OOD) scenarios. In our task, specifically, when the leader presents unseen motion patterns, the follower may displace to response without proper foot steps thus yielding skating artifacts. To better adapt GPT models to this challenge, we employ an off-policy reinforcement learning (RL) to finetune the network. Unlike the supervised training stage, there is no ground truth labels to regress in RL stage. As shown in Figure 5, for RL, the inputs of GPT model are code sequences $\{\hat{z}\}$ generated by the GPT itself on OOD conditions, either using the current network weight or the past ones. The network weight is then optimized according to a loss based on human-defined rewards.

From an RL standpoint, the act of predicting the next token at step $t$ is construed as an *action* $a_t$, which involves selecting an appropriate token $z_{t+1}$ from the dictionary. Concurrently, the GPT model with weight $\theta$ is considered a policy network $\pi_\theta$ that predicts a probability distribution $\pi_\theta(\cdot|s_t)$ for each candidate $k \in \mathcal{Z}$ at step $t$, with the state $s_t$ perceived as the preceding sequence $\{z_0, z_1, ..., z_t\}$. At present, a majority of the publicly documented RL implementations (Ouyang et al., 2022; Siyao et al., 2022) for GPT employ on-policy actor-critic (AC) strategy, where the update of the policy network hinges on the sequences generated by the *current* GPT weight. Basically, AC uses a loss function as follows:

$$
\mathcal{L}^{on}_{AC}(\theta) = \sum_{t=0}^{T-1} -\log\left(\pi_\theta(\hat{a}^\theta_t|\hat{s}^\theta_t)\right) \cdot A,
\tag{4}
$$

where $\{\hat{a}^\theta_t\}$ denotes the trajectory sampled by the *immediate* GPT weight $\theta$. The term $A$ represents the "advantage", which reveals the increment after taking action $\hat{a}^\theta_t$ (see Appendix C.1). A positive advantage increases the probability $\pi(\hat{a}^\theta_t|\hat{s}^\theta_t)$ of the currently sampled action $\hat{a}^\theta_t$, whereas a negative advantage reduces it. This indiscriminate mechanism of increasing or decreasing hinders the ability of RL to reuse past data. For example, when the advantage of a previous sampling $(\hat{s}^{\theta_{old}}_t, \hat{a}^{\theta_{old}}_t)$ is negative, the on-policy loss $\mathcal{L}^{on}_{AC}$ will persist in pushing associated values to decrease, regardless of the current probability already being very small. Such misalignment is counter-logical and potentially harmful to network training due to the misleading optimization target.

To address this issue, we extend the RL on GPT off policy by establishing an *explicit optimization target* for the probability $\pi_\theta(\hat{a}_t|\hat{s}_t)$. Drawing from ideas in energy-based policy modeling (Haarnoja et al., 2017; 2018), an equivalence can be established between the policy probability and the expected reward income, implemented by a monotonic mapping $\sigma : \mathbb{R} \to [0, 1]$. This relationship can be formulated as $\pi_\theta(a|s) = \sigma(Q(s, a))$, where the $Q$ value represents the expected total future income after executing action $a$ at state $s$. In light of this equivalence, if an action $a$ results in greater future income $Q(s, a)$, it should hold a larger probability $\sigma(Q(s, a))$ of being selected, which leads to an explicit optimization reference to $\pi_\theta(a|s)$. With this concept as our foundation, we construct a novel learning strategy using a loss function defined as

$$\mathcal{L}_{RL}^{\text{off}}(\theta) = \sum_{t=0}^{T-1} -\log\left(1 - \text{abs}\left[\pi_\theta\left(\hat{a}_t|\hat{s}_t\right) - \sigma\left(Q\left(\hat{s}_t, \hat{a}_t\right)\right)\right]\right). \tag{5}$$

The proposed learning strategy is off-policy since $\mathcal{L}_{RL}^{\text{off}}$ can be reused on past data to guide the GPT to learn a specific probability of $\hat{a}_t$. In practice, we set $\sigma = \text{sigmoid}(\alpha * Q + \beta)$ and approximate $Q$ by two-step accumulation of rewards on sampled trajectory as $\hat{Q} \approx r(\hat{s}_t, \hat{a}_t) + \gamma r(\hat{s}_{t+1}, \hat{a}_{t+1})$. Detailed algorithms pipeline is shown in the supplementary file.

**Step-wise Rewards in Dance Accompaniment.** In this implementation, we specifically address the skating artifacts caused by the inconsistency between predicted translation and the lower body movement. To this end, we design a step-wise reward for each predicted token. Specifically, we train an additional velocity decoding branch, namely $D_V$, for lower-body motion VQ-VAE in an unsupervised manner, as per implementation by Siyao et al. (2022), to decode the predicted lower body sequence $z^{down\heartsuit}$ to velocity $V$ by $V = D_V(z^{down\heartsuit})$. We then use $V$ as a reference to judge whether the follower's global position $q^\heartsuit$, calculated from translation, synchronizes with the lower body movement by computing a difference $\delta$ as $\delta = \sum_{u=0}^{d-1} \|\dot{q}^\heartsuit{}_{t\cdot d+u} - V_{t\cdot d+u}\|/d$, where $\dot{q}^\heartsuit$ is the first derivative approximated by subtraction between adjacent items. The reward $r_t^{down}$ for lower body at step $t$ is then defined as

$$r_t^{down} = \begin{cases} 1, & \text{if } \delta < \text{threshold}, \\ -\eta \cdot \delta, & \text{otherwise}, \end{cases} \tag{6}$$

where $\eta$ is a parameter to control the punishment magnitude. $\eta$ is set to 100 and the threshold is 0.03 in our experiment. Rewards for other components ($up$, $lhand$, $rhand$ and $tr$) are consistent to be 1.

## 4 EXPERIMENTS

Detailed descriptions and implementations of the VQ-VAE and GPT are in Appendix B.

**Evaluation Metrics.** We apply a series of quantitative metrics to evaluate the generated follower's movement in three distinct aspects: **(1)** the inherent quality of the follower's motion, independent of its interaction with the leader, **(2)** the follower's interaction with the leader, and **(3)** the alignment of the dance with the background music. For **(1)**, we adopt metrics used in the solo dance benchmark AIST++ (Li et al., 2021b). Specifically, we compute the Fréchet Inception Distance (FID) between the generated follower's motion and the real dance in the whole `DD100` dataset on kinematic (denoted as "$k$") (Onuma et al., 2008) and graphical (denoted as "$g$") (Müller et al., 2005) features, and compute the standard deviation between features of the generated follower sequences as the diversities $\text{Div}_k$ and $\text{Div}_g$ that indicate the difference among those sequences. As to **(2)**, the interactive quality, we first extract a *cross-distance* ($cd$) feature from the duet dance sequences. Specifically, for each frame, we calculate the pairwise distances between ten joints' positions, including the pelvis, both knees, feet, shoulders, the head, and two twists, of the leader and those of the follower to obtain a 100-dimension ($10 \times 10$) features. Then, we use these distances as an interactive feature and compute the $\text{FID}_{cd}$ and $\text{Div}_{cd}$. Meanwhile, we compute contact frequency (CF) value to explicitly explore the strength of interaction. CF is defined as the ratio of the number of frames in which two dancers are in physical contact, where physical contact is defined as two SMPL-X models having a minimum absolute mesh distance below a 2-*cm* threshold. Finally, we define Beat Echo Degree (BED) to evaluate the consistency of dynamic rhythms of two dancers:

$$\frac{1}{|B^l|} \sum_{t^l \in B^l} \exp\left\{-\frac{\min_{t^f \in B^f} \|t^l - t^f\|^2}{2\sigma^2}\right\}, \tag{7}$$

Table 2: **Quantitative benchmark for dance accompaniment.** The first place and runner-up are highlighted in bold and underlined, respectively. S denotes a solo dance generation model that does not take the leader into condition, while D denotes that one does. *Since solo dance has no interaction, the cross-distance between two agents are completely irregular, making the diversity particularly high.

| Method | Solo Metrics | | | | Interactive Metrics | | | | Rhythmic |
| --- | --- | --- | --- | --- | --- | --- | --- | --- | --- |
| | $\text{FID}_k(\downarrow)$ | $\text{FID}_g(\downarrow)$ | $\text{Div}_k(\uparrow)$ | $\text{Div}_g(\uparrow)$ | $\text{FID}_{cd}(\downarrow)$ | $\text{Div}_{cd}(\uparrow)$ | CF(%) | BED($\uparrow$) | BAS($\uparrow$) |
| Ground Truth | 6.56 | 6.37 | 11.31 | 7.61 | 3.41 | 12.35 | 74.25 | 0.5308 | 0.1839 |
| S _Bailando_ (Siyao et al., 2022) | 78.52 | 36.19 | 11.15 | 7.92 | 6643.31 | 52.50* | 7.13 | 0.1831 | 0.1930 |
| S EDGE (Tseng et al., 2023) | 69.14 | 44.58 | 8.62 | 6.35 | 5894.45 | 60.62* | 6.82 | 0.1822 | 0.1875 |
| S _Duolando_ w/o. RL _tr_ IC | **12.53** | **24.17** | 10.51 | **9.42** | 4803.20 | 42.72* | 7.04 | 0.1826 | 0.1852 |
| D _Duolando_ w/o. RL _tr_ | 62.29 | 27.95 | 13.16 | 8.53 | 7970.19 | 54.53* | 7.76 | 0.2194 | 0.2002 |
| D _Duolando_ w/o. RL | 106.72 | 34.10 | **13.88** | 7.03 | 21.68 | 9.33 | **57.43** | 0.2795 | **0.2193** |
| D _**Duolando**_ | 25.30 | 33.52 | 10.92 | 7.97 | **9.97** | **14.02** | 52.36 | **0.2858** | 0.2046 |

where $B^l = \{t^l\}$ and $B^f = \{t^f\}$ represent the timing of beats in the leader and follower movements, respectively, while the motion beat time is calculated by finding the local minimum time of the motion velocity; $\sigma$ is a normalization parameter ($\sigma = 3$ in our experiments). The BED takes a similar formulation as the Beat-Align Score (BAS) (Siyao et al., 2022), where the reference base is substituted with leader's dynamic beats instead of the music's rhythmic beats. For **(3)**, we adopt the BAS defined by Siyao et al. (2022), to assess the correspondence between the generated motion and the rhythm of the background music.

**Baseline Setup.** In addition to our proposed _Duolando_, we evaluate two state-of-the-art solo dance generation methods, _Bailando_ (Siyao et al., 2022) and EDGE (Tseng et al., 2023), to discern the differences between solo dance and interactive duet dance. Considering the absence of existing dance accompaniment methods conditioned on both music and a leader's motion, we also perform ablation studies using various _Duolando_ variants. Specifically, we investigate the effectiveness of reinforcement learning (RL), relative translation ($tr$), and interaction coordination (IC), through the evaluation of variants "w/o. RL", "w/o. RL $tr$", and "w/o. RL $tr$ IC", respectively. For the variant without $tr$, we delete $z^{tr}$ from the input of GPT and predict the translation via the velocity decoding branch $D_V$, which is trained in an unsupervised manner, as detailed in Section 3.3 and (Siyao et al., 2022). In the case of "w/o. RL $tr$ IC", we further simplify the model's input to $\boldsymbol{m}$, $z^{up\heartsuit}$, $z^{down\heartsuit}$, $z^{lhand\heartsuit}$, and $z^{rhand\heartsuit}$, excluding the influence of the leading dancer, to observe the effectiveness of interactive coordination in dance accompaniment. Along with these methods, we include the scores of the ground truth test data as reference points. The quantitative benchmark is provided in Table 2.

**Analysis.** Given that the methods presented in Table 2 are organized in an order with incremental addition of modules, we conduct a detailed pairwise analysis of each successive pair. Upon examination of the first two solo dance models, _Bailando_ achieves $\text{FID}_k$ and $\text{FID}_g$ scores of 78.52 and 36.19, respectively, while those of EDGE are 69.14 and 44.58. In contrast, the model "_Duolando_ w/o. RL $tr$ IC" significantly improves upon these solo metrics, with scores increasing by 65.99 (84%) and 56.61 (72%) respectively. Compared to _Bailando_, this variant of _Duolando_ includes a look-ahead (LA) mechanism, which allows for the generation of more fluent and continuous dance movements over long durations, contributing to the significant improvement of the FID scores. However, since all solo dance methods lack the condition of a leader, they do not exhibit a noticeable change in poor interactive values. By taking the leader's movement into account and using the proposed interactive coordination (IC), "_Duolando_ (w/o. RL $tr$)" achieves a 20% improvement on BED, increasing from 0.1826 to 0.2194, which can be attributed to better synchronization with the leader's dynamics. Nonetheless, when compared with "_Duolando_ w/o. RL $tr$ IC", "Duolando w/o. RL $tr$" still shows suboptimal performance in terms of $\text{FID}_{cd}$ and $\text{Div}_{cd}$. This is because the $cd$ feature is strongly affected by the relative translation between the two dancers. While the virtual follower can respond to the leader's motion pattern, it still fails with positioning itself in a reasonable location based on unsupervised velocity prediction. Moreover, the low CF value of 7.76% indicates that the generated follower rarely makes contact with the leader, contradicting the fundamental requirements of dance accompaniment. With the introduction of explicit relative translation ($tr$) prediction via interactive coordination, the $\text{FID}_{cd}$ of "_Duolando_ w/o. RL" reduces drastically from 7970.19 to 21.68, aligning much more closer with the ground truth. Simultaneously, the CF value increases to 57.43% (a 7-fold increment), and BED is raised to 0.27955 (27% $\uparrow$), indicating a higher level of interaction with

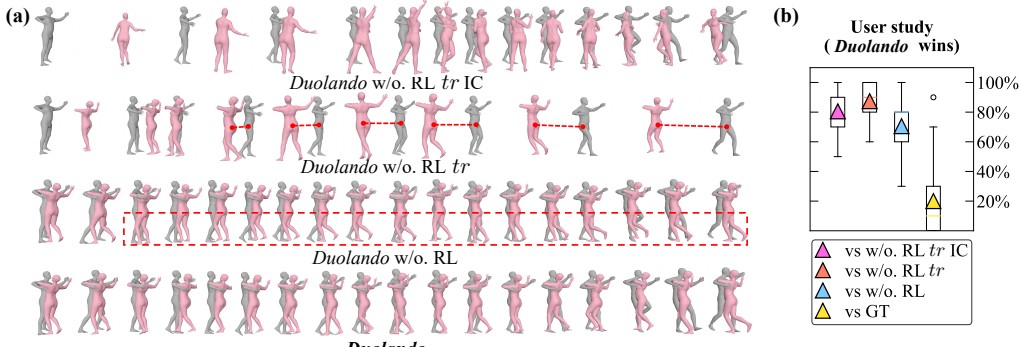

Figure 6: **Qualitative results (a) and user study (b).** In qualitative results, conditioning leader is colored in gray while generated followers are in red. In boxplot of user study, triangles and colored lines are mean and median values, respectively. Circles are outliers beyond $1.5\times$ interquartile range ($3\sigma$ in normal dist.).

the leader. However, the explicitly predicted translation can sometimes clash with the generated movements, deteriorating the quality of the dance generated. Upon closer examination, compared to "*Duolando* w/o. RL $tr$", the FID$_k$ value of "*Duolando* w/o. RL" drops significantly from $62.29$ to $106.72$ ($71\% \uparrow$). Given that the kinetic features (Onuma et al., 2008) are computed based on motion velocity, they are particularly susceptible to unreasonable global shifts like the so-called "skating artifacts". This issue is resolved by the proposed reinforcement learning mechanism, enabling the full ***Duolando*** model to show a $81.42$ ($76\%$) improvement on FID$_k$ compared to the variant without RL. Moreover, the reinforcement learning mechanism contributes to an improvement in interactive quality ($11.71$, $54\% \downarrow$), diversity ($4.69$, $50\% \uparrow$) and motion alignment ($0.063$, $22\% \uparrow$), demonstrating its effectiveness in enhancing the performance of the full ***Duolando*** model. In conclusion, the improvements of different metrics attest to the effectiveness of each proposed module, underscoring their potential in producing more fluent, interactive, and diverse dance movements.

**Qualitative Comparisons.** To clearly demonstrate the effectiveness of each proposed module, we provide several visualizations of results produced by different variations in Figure 6. In the case of "*Duolando* w/o. RL $tr$ IC", the model lacks a condition for the leader, resulting in the virtual follower operating independently and not responding to the "dance hold" (the pose holding arms) initiated by the leader. Conversely, the follower in "*Duolando* w/o. RL $tr$" can pose responsively, but she still fails to accurately follow the leader's displacement, causing a unreasonable distance between the dancers. The relative translation module allows the two dancers to interact closely in "Duolando w/o. RL". However, as highlighted in the red dotted box, the follower's legs do not move appropriately when shifting with the leader, leading to a noticeable skating artifact. With the integration of reinforcement learning, the follower can alternate footwork, presenting a proper displacement of steps backwards.

**User Study.** To explore subjective judgments of quality, we conduct a user study wherein ***Duolando*** is compared against each of its variants individually. As both *Bailando* and "*Duolando* w/o. RL $tr$ IC" are solo dance generation frameworks, we only conduct this study exclusively on the latter model, which demonstrates higher performance quantitatively. Specifically, we recruit 15 participants, while for each participant, we randomly display 40 pairs of dance clips, with each pair comprising one ***Duolando*** result and one result from the comparison method. We then ask to indicate *which one dances better in response to the leader and the music*. As depicted in Figure 6(b), the full ***Duolando*** model outperforms the "w/o. RL $tr$ IC", w/o. RL $tr$", and "w/o, RL" variants in $80\%$, $83\%$, and $62\%$ of comparisons respectively, demonstrating the significance of each proposed module. However, when compared to the ground truth, ***Duolando*** only surpasses it around $15\%$ of the time, indicating that dance accompaniment remains a challenging and open task for future works.

## 5 CONCLUSION

We introduce a new task named dance accompaniment. To support this task, we first collect a large-scale duet dance dataset `DD100`, and propose a baseline framework named Duolando, which develops a follower GPT with off-policy reinforcement learning. We establish a benchmark with several metrics to evaluate the dance quality, interaction, and alignment with music. Additionally, `DD100` has the potential to support more multi-modal human-human interaction tasks.

**Ethics Statement** The task we're proposing - dance accompaniment - holds significant potential for enhancing a multitude of VR/AR applications, including the prospect of virtually dancing alongside AI. Furthermore, research into human-human interaction could substantially aid in improving the immersive and interactive experiences offered by VR games. However, such applications or games have a potential risk in the future that users might be addicted to interacting with virtual agent instead of real-world social events, once the response of the virtual agent is developed to be charming and real-looking enough. Meanwhile, the research on real-looking motion generation in response to human may also lead to AI fraud, where the existing verifying methods could be more difficult to judge whether it is fake based on response.

The data collection process (including contracts with actors) in our research was conducted ethically. The data will be released without containing any personally identifiable information. Some audio tracks in our dataset come from copyrighted musics. The inclusion of these tracks should be regarded as 'fair use' provisions due to three reasons. (1) The music segments used are modified (shortened, and partially slowed down), deviating from their original form. (2) The use of these tracks is solely for model training/testing the model rather than entertainment. (3) The academic nature of this work and the absence of public searchability (we won't provide music info beyond the audio signals) avoid potential commercial impact.

**Acknowledgement** This research is supported by the National Research Foundation, Singapore under its AI Singapore Programme (AISG Award No: AISG-PhD/2021-01-031[T]). This study is supported under the RIE2020 Industry Alignment Fund Industry Collaboration Projects (IAF-ICP) Funding Initiative, as well as cash and in-kind contribution from the industry partner(s). It is also supported by Singapore MOE AcRF Tier 2 (MOE-T2EP20221-0011) and AcRF Tier 2 (MOE-T2EP20221-0012).

We sincerely thank Mingyang Song, the PM in SenseTime, for helping kick off this project and thank Judy for organizing the dancers. We also appreciate Lai Jiang, Zhijie Cao, Tinghao Liu and Quan Wang for their significant help during data pre-processing. Siyao is grateful to Mr. Zhuo Sun and Dr. Chen Qian for their indispensable supports. The Mocap data are collected through services of QINGMU TECH LTD., Shanghai.

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

## A  RELATED WORK

Recent years have witnessed numerous studies on conditional human motion synthesis. One one hand, it can be classified according to conditions, such as text-to-motion (Petrovich et al., 2021; Guo et al., 2022; Tevet et al., 2022; Hong et al., 2022; Petrovich et al., 2022; Athanasiou et al., 2022; Zhang et al., 2022; 2023a;b), voice-to-gesture (Ng et al., 2022; Li et al., 2021a; Ao et al., 2022; Yang et al., 2023a) and music-to-dance (Li et al., 2021b; 2022; Siyao et al., 2022; Le et al., 2023). On the other hand, it can be categorized based on the type of interaction, including single-human motion synthesis (Huang et al., 2021; Li et al., 2021b; Siyao et al., 2022; Zhang et al., 2022; 2023b; Gong et al., 2023; Jiang et al., 2023), group human motion synthesis (Le et al., 2023; Vendrow et al., 2022), human-scene interaction (Zhao et al., 2022; Yi et al., 2022; 2023), human-object interaction (Jain & Liu, 2009; Ghosh et al., 2023; Li et al., 2023), and human-human interaction (cmu; Men et al., 2022; Shafir et al., 2023; Liang et al., 2023).

### A.1  MUSIC-TO-DANCE

**Datasets.** Music-to-dance represents a crucial category within the realm of conditional human motion synthesis, where music serves as the input to generate a sequence of human body movements. Previous studies have made significant strides in this field by developing several datasets (Tang et al., 2018; Mahmood et al., 2019; Li et al., 2021b; Valle-Pérez et al., 2021; Cai et al., 2021; Li et al., 2022; Cai et al., 2022; Yang et al., 2023b; Le et al., 2023) either compiled through motion capture (MoCap) systems or reconstructed from videos using automated algorithms. AIST++(Li et al., 2021b) is a

dataset derived from multi-camera videos spanning 10 dance genres and 30 subjects, compiling a total of 1408 sequences or 5.2 hours of content. AIOZ-GDANCE(Le et al., 2023) consists of 16.7 hours of paired music and 3D motion gathered from in-the-wild videos, featuring 7 dance styles and 16 music genres. Despite the advantage of an automated approach in terms of large-scale dataset collection, the quality of reconstruction is often compromised, particularly when occlusions are present or human poses are complex. To capture high-quality motions, professional motion capture equipment is utilized. Dance Melody (Tang et al., 2018) employs the MoCap of 3D skeletons from dancers, establishing a music-to-dance dataset with four types of dance. PMSD (Valle-Pérez et al., 2021) uses an Optitrack system comprising 17 cameras to collect 142 minutes of casual dancing and 44 minutes of street dance. Notably, DanceFormer (Li et al., 2022) employs animators to manually create high-quality dancing data. It took approximately 18 months for the animators to produce the 300 dance animations, which underscores the scalability challenges inherent in this approach. However, these datasets do not incorporate strong interactions between individuals, rendering them unsuitable for either dance accompaniment or duet dance generation tasks.

**Existing Methods.** With the rapid development of datasets and benchmarks, numerous new methods have emerged in this field. To generates 3D dance with input music, Dance Revolution (Huang et al., 2021) formalizes music-conditioned dance generation as a sequence-to-sequence learning problem and devise a seq2seq architecture. FACT (Li et al., 2021b) incorporates a deep cross-modal transformer block with full-attention. *Bailando* (Siyao et al., 2022) proposes a two-stage procedure, where the first stage applies a VQ-VAE to quantize dancing-style poses and the latter stage employs an on-policy actor-crtic GPT to generate dancing motions, improving temporal coherency with music. GDanceR (Le et al., 2023) is recently proposed to produce coherent group dance, with a transformer music encoder and a group motion generator. The existing methods, however, focus on either synthesizing solo dance motion or simultaneously generating multiple agents with weak interactions, which is not suitable for the requirement to respond to preconditioned lead dancing motion.

In contrast to previous datasets and methods, we propose a new task called dance accompaniment that requires conditioning on both music and the leader's dance movements. We collect a large-scale and diverse duet dance dataset named DD100, which consists of both dancing sequences and the music signals, and develop a baseline method on top of it, along with evaluation protocols.

## A.2 HUMAN-HUMAN INTERACTION

**Datasets.** As the quality of single-person motion generation improves, researchers are shifting their focus to the human-human interaction task. An early attempt in this direction is CMU-MoCap (cmu), which collects weak interactions like handshaking using motion capture equipment. Since then, various multi-person motion datasets have been developed, including NTU-RGBD (Liu et al., 2020), You2Me (Ng et al., 2020), and UMPM (Van der Aa et al., 2011). However, while these datasets contain multi-person motion data, they are limited in interaction type and strength. A recent dataset, InterHuman (Liang et al., 2023), also captures interactions through MoccCap, and labels the motions with textual descriptions. However, InterHuman does not provide music as a multi-modality condition, which is not comprehensive enough for the task of dance accompaniment.

**Existing Methods.** With the availability of these datasets, several methods have been proposed for generating human-human interaction motions. (Men et al., 2022) propose a seq2seq GAN system that synthesizes the reactive motion of a character given the active motion from another character. ComMDM (Shafir et al., 2023) devises a communication block to infuse interaction between two generated motions, using a pretrained diffusion-based model as a generative prior. InterGen (Liang et al., 2023) tailors the motion diffusion model to a two-person interaction setting, via two cooperative transformer-based denoisers with mutual attention mechanism.

In contrast to these human-human interaction datasets and approaches, our dance accompaniment introduces the music modality and requires adherence to both the rhythm of the music and the motion from another person. Moreover, due to the unique nature of dance, it involves long-term strong interaction between dancers.

# B  MORE DETAILS

In this section we introduce statistics of the motion speed in different dance types, the detailed network structures and training process of ***Duolando***.

## B.1  SPEED STATISTICS

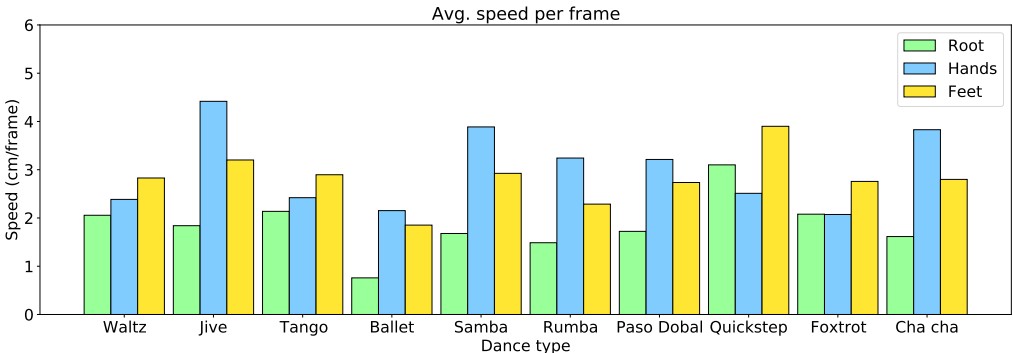

Figure 7: **Statistics of average speed values (cm per frame) for different dances.** We present the speed values of three kinds of joints: root (pelvis), hands and feet.

We calculated the average speed for each dance type by computing the average difference in 3D coordinates between frames for the root (pelvis), hands, and feet, as shown in Figure 7. Typically, Ballet involves many spinning in place, resulting in a low overall global (root) speed, while Quickstep, known for its lightness and speed, exhibit a higher overall velocity. Meanwhile, some dances with relatively low overall movement speed but expressive hand movements (like Jive and Cha Cha), contrast with those that have smaller hand variations and primarily rely on footwork (such as Waltz and Foxtrot).

## B.2  NETWORK STRUCTURES

**VQ-VAEs**. The encoder and decoders in Section 3.1 of the main paper are complied as 1D temporal CNNs. The detailed structures are shown in Table 3. The encoder is designed to downsample the 3D joint sequence 8 times into the encoding features, and therefore the learned quantized features are aware of contextual poses in the dance sequence.

Separate VQVAEs are trained for the compositional upper-and-lower half bodies, respectively. The joint number $J$ of the upper half body is 15 and that of the lower is 9, while $J = 1$ in the global-velocity branch.

**GPT**. We follow the fundamental structure of minGPT by Karpathy (2020). Specifically, the we employ 12 Transformer layers with the block size $T$ of 30 and feature dimension of 768, while the LAT consists of three Transformer layers with the look-ahead length $L$ of 29 (around 4 seconds).

The music features $m$ are extracted from the audio signal using the public audio processing toolbox Librosa (McFee et al., 2015), including mel frequency cepstral coefficients (MFCC), MFCC delta, constant-Q chromagram, onset strength and onset beat, which are 54-dimension in total, and are mapped to the same dimension of GPT via a learned linear layer.

## B.3  TRAINING HYPER-PARAMETERS

In terms of hyper-parameters, we set the codebook capacity $K$ to 512 for all quantized items ($z^{up}$, $z^{down}$, $z^{lhand}$, $z^{rhand}$, and $z^{tr}$). During VQ-VAE training, we segment motion sequences into 4-second ($T = 120$) slices with a batch size of 64. The encoder's temporal downsampling rate $d$ is set to 4, and the commitment trade-off $\lambda$ is 0.1. We adopt a learning rate of $3 \times 10^{-5}$ to train the VQ-VAE for 500 epochs, with a decay of 0.1 after both the 200th and 300th epochs. Throughout the supervised training state of GPT, we adopt cross-entropy loss with a learning rate of $10^{-4}$ for

Table 3: **Architectures of VQ-VAE encoder and decoders. (Section 3.1)**. "Conv" and "TransConv" represent 1D temporal the convolution and transpose-convolution operations, respectively, and their arguments represent the input channel number, the output channel number, the kernel size, the convolution stride, the padding size on both ends of input data, and the dilation number in turn. **RB** denotes Residual Block. $J$ denotes the 3D joint number.

|  | **Residual Block** |
|---|---|
|  | **Input: 0**; **Argument:** $p, d$ |
| **1** | ReLU, Conv(512, 512, 3, 1, $p$, $d$) |
| **2** | ReLU, Conv(512, 512, 1, 1, 0, 1) |
|  | **Output: 0 + 2** |

|  | **Encoder** $E$ |
|---|---|
|  | **Input: 0, Argument:** $J$ |
| **1** | Conv($J \times 3$, 512, 4, 2, 1, 1) |
| **2** | **RB**($p = 1, d = 1$) |
| **3** | Conv(512, 512, 4, 2, 1, 1) |
| **4** | **RB**($p = 3, d = 3$) |
| **5** | Conv(512, 512, 4, 2, 1, 1) |
| **6** | **RB**($p = 9, d = 9$) |
|  | **Output: 6** |

|  | $D_P$ |
|---|---|
|  | **Input: 0, Argument:** $J$ |
| **1** | **RB**($p = 9, d = 9$) |
| **2** | TransConv(512, 512, 4, 2, 1, 1) |
| **3** | **RB**($p = 3, d = 3$) |
| **4** | TransConv(512, 512, 4, 2, 1, 1) |
| **5** | **RB**($p = 3, d = 3$) |
| **6** | TransConv(512, 512, 4, 2, 1, 1) |
| **7** | Conv(512, $J \times 3$, 3, 1, 1, 1) |
|  | **Output: 7** |

|  | $D_M$ |
|---|---|
|  | **Input: 0, Argument:** $J$ |
| **1** | **RB**($p = 9, d = 9$) |
| **2** | TransConv(512, 512, 4, 2, 1, 1) |
| **3** | **RB**($p = 3, d = 3$) |
| **4** | TransConv(512, 512, 4, 2, 1, 1) |
| **5** | **RB**($p = 3, d = 3$) |
| **6** | TransConv(512, 512, 4, 2, 1, 1) |
| **7** | Conv(512, $J \times 9$, 3, 1, 1, 1) |
|  | **Output: 7** |

500 epochs. For the reinforcement learning stage, we apply $\mathcal{L}_{RL}^{\text{off}}$ learning rate of $3 \times 10^{-5}$ for 50 epochs. The OOD dataset $\hat{\mathcal{D}}$ is compiled on the sequences *generated* by GPT conditioned on the music and leader motion in test set, without using any ground-truth follower information. During the entire training process, we employ the Adam optimizer (Kingma & Ba, 2014) with $\beta_1 = 0.9$ and $\beta_2 = 0.99$. The training is conducted on four NVIDIA Tesla V100 GPUs, taking approximately seven days in total.

## C  REINFORCEMENT LEARNING DETAILS

### C.1  DERIVATION TO ON-POLICY ACTOR-CRITIC LOSS

Here we derive the on-policy actor-critic loss $\mathcal{L}_{AC}$ from the initial reinforcement learning objective. The derivation is mainly based on the instruction of Levine.

From an RL standpoint, GPT predicting the next token at step $t$ is construed as an *action* $a_t$, which involves selecting an appropriate token $z_{t+1}$ from the dictionary. As such, the cross-entropy loss in the supervised stage can be reformulated as

$$\mathcal{L}_{ST}(\theta) = \sum_{t=0}^{T-1} -\log\left(\pi_\theta(a_t|s_t)\right). \tag{8}$$

For RL, the object $J(\theta)$ is to maximize the total income $\mathbb{E}_{\tau \sim \pi_\theta(\tau)}[r(\tau)]$, while it can be reframed as

$$J(\theta) = \mathbb{E}_{\tau \sim \pi_\theta(\tau)}[r(\tau)] = \sum_{\text{all } \tau} \pi_\theta(\tau) r(\tau), \tag{9}$$

where $\theta$ denotes the weights of the policy making network, $\tau$ represents a string of actions $\tau = \{a_t\}_{t=0}^{T-1}$ and $\pi_\theta(\tau)$ is the probability $\prod_{t=0}^{T-1} \pi_\theta(a_t|s_t)$ that the policy network predicts to take such trajectory; $r(\tau)$ is the total income taking trajectory $\tau$ as $\sum_{t=0}^{T-1} r(a_t, s_t)$.

---

**Algorithm 1** Off-Policy RL in *Duolando*

---

**Require:** pertained GPT weights $\theta$, OOD conditions $\{(\hat{m}, \hat{z})\}$, Epoch number $\mathcal{E}$

1: $\epsilon \leftarrow 0$
2: $\hat{\mathcal{D}} \leftarrow \varnothing$
3: **while** $\epsilon < \mathcal{E}$ **do**
4:     generate sequence $\{\hat{z}^\epsilon, \hat{tr}^\epsilon\} \leftarrow \text{GPT}_\theta(\hat{m}, \hat{z})$
5:     $\hat{\mathcal{D}} \leftarrow \hat{\mathcal{D}} \cup \{(\hat{m}, \hat{z}, \hat{z}^\epsilon, \hat{tr}^\epsilon)\}$                      ▷ Reuse past generated data
6:     **for** $(\hat{m}, \hat{z}, \hat{z}, \hat{tr}) \in \hat{\mathcal{D}}$ **do**
7:         $\hat{s} \leftarrow (\hat{m}, \hat{z}, \hat{z}_{0...T-1}, \hat{tr}_{0...T-1})$
8:         $\hat{a} \leftarrow (\hat{z}_{1...T}, \hat{tr}_{1...T})$
9:         Compute $r_{1...T}$ as described in Sec. 3.3
10:        Compute $\mathcal{L}_{RL}^{\text{off}}$ by Eq. (4) (main paper)
11:        $\theta \leftarrow \theta - \nabla_\theta L_{RL}^{\text{off}}$
12:     **end for**
13:     $\epsilon \leftarrow \epsilon + 1$
14: **end while**

---

One approach to maximize $J$ is to optimize the network weight $\theta$ along the gradient $\nabla_\theta J$ as $\theta \leftarrow \theta + \alpha \nabla_\theta J$. Since

$$\nabla_\theta \pi_\theta = \pi_\theta \frac{\nabla_\theta \pi_\theta}{\pi_\theta} = \pi_\theta \nabla_\theta \log \pi_\theta, \tag{10}$$

we can rewrite $\nabla_\theta J$ into

$$\begin{aligned}
\nabla_\theta J &= \sum_\tau \nabla_\theta \pi_\theta(\tau) r(\tau) \\
&= \sum_\tau \pi_\theta \nabla_\theta(\tau) \log \pi_\theta(\tau) r(\tau) \\
&= \mathbb{E}_{\tau \sim \pi_\theta(\tau)} \left[ \nabla_\theta \log \pi_\theta(\tau) r(\tau) \right].
\end{aligned} \tag{11}$$

Since $\pi_\theta(\tau) = \prod_{t=0}^{T-1} \pi_\theta(a_t | s_t)$, we have

$$\nabla_\theta \log \pi_\theta(\tau) = \sum_{t=0}^{T-1} \nabla_\theta \log \pi_\theta(a_t | s_t). \tag{12}$$

Hence,

$$\begin{aligned}
\nabla_\theta J &= \mathbb{E}_{\tau \sim \pi_\theta(\tau)} \left[ \nabla_\theta \log \pi_\theta(\tau) r(\tau) \right] \\
&= \mathbb{E}_{\tau \sim \pi_\theta(\tau)} \left[ \left( \sum_{t=0}^{T-1} \nabla_\theta \log \pi_\theta(a_t | s_t) \right) \left( \sum_{t=0}^{T-1} r(a_t, s_t) \right) \right] \\
&= \mathbb{E}_{\tau \sim \pi_\theta(\tau)} \left[ \sum_{t=0}^{T-1} \nabla_\theta \log \pi_\theta(a_t | s_t) \left( \sum_{u=0}^{T-1} r(a_u, s_u) \right) \right].
\end{aligned} \tag{13}$$

For on-policy reinforcement learning, $\nabla_\theta J$ is estimated on simultaneously sampled sectional trajectories $\{(\hat{a}_t^\theta, \hat{s}_t^\theta)\}$, such that the equation above is approximated to be

$$\nabla_\theta J \approx \sum_{t=0}^{T-1} \nabla_\theta \log \pi_\theta(\hat{a}_t^\theta, \hat{s}_t^\theta) \left( \sum_{u=0}^{T-1} r(\hat{a}_u^\theta, \hat{s}_u^\theta) \right). \tag{14}$$

Note that the optimization of the policy making network for policy $(a_t, s_t)$ is not expected to be influenced by the past trajectories, *i.e.*, the rewards before $t$. Therefore, Equation (14) is reframed as

$$\nabla_\theta J \approx \sum_{t=0}^{T-1} \nabla_\theta \log \pi_\theta(\hat{a}_t^\theta, \hat{s}_t^\theta) \left( \sum_{u=t}^{T-1} r(\hat{a}_u^\theta, \hat{s}_u^\theta) \right), \tag{15}$$

where $\sum_{u=t}^{T-1} r(\hat{a}_u^\theta, \hat{s}_u^\theta)$ is the expected total future income after taking $\hat{a}_u^\theta$ under state $\hat{s}_u^\theta$, or to say, the expected "reward to go" which is formally named as the Q-value $Q(\hat{a}_t^\theta, \hat{s}_t^\theta)$.

Table 4: **Ablation studies for music input.** ⊠ represents random music input.

| Method | Solo Metrics | | | | Interactive Metrics | | | | Rhythmic |
|---|---|---|---|---|---|---|---|---|---|
| | $\mathrm{FID}_k(\downarrow)$ | $\mathrm{FID}_g(\downarrow)$ | $\mathrm{Div}_k(\uparrow)$ | $\mathrm{Div}_g(\uparrow)$ | $\mathrm{FID}_{cd}(\downarrow)$ | $\mathrm{Div}_{cd}(\uparrow)$ | CF(%) | BED(↑) | BAS(↑) |
| Ground Truth | 6.56 | 6.37 | 11.31 | 7.61 | 3.41 | 12.35 | 74.25 | 0.5308 | 0.1842 |
| Ground Truth (⊠) | 6.56 | 6.37 | 11.31 | 7.61 | 3.41 | 12.35 | 74.25 | 0.5308 | 0.1760 |
| *Duolando* | 25.30 | 33.52 | 10.92 | 7.97 | 9.97 | 14.02 | 52.36 | 0.2858 | 0.2046 |
| *Duolando* (⊠) | 23.73 | 35.91 | 10.95 | 8.36 | 7.77 | 13.96 | 51.77 | 0.2582 | 0.2072 |

To avoid the bias of rewards, *e.g.*, all rewards are positive, the Q-value item in Equation (15) is normalized by an expected "reward to go" on state $s_t$, *i.e.*, the critic value $V_t = \mathbb{E}_{a_t}[Q(a_t, s_t)]$, such that

$$\nabla_\theta J \approx \sum_{t=0}^{T-1} \nabla_\theta \log \pi_\theta(\hat{a}_t^\theta, \hat{s}_t^\theta) \left( Q(\hat{a}_t^\theta, \hat{s}_t^\theta) - V(\hat{s}_t^\theta) \right)$$

$$= \sum_{t=0}^{T-1} \nabla_\theta \log \pi_\theta(\hat{a}_t^\theta, \hat{s}_t^\theta) \left( r(\hat{a}_t^\theta, \hat{s}_t^\theta) + V(\hat{s}_{t+1}^\theta) - V(\hat{s}_t^\theta) \right),$$

(16)

where $r(\hat{a}_t^\theta, \hat{s}_t^\theta) + V(\hat{s}_{t+1}^\theta) - V(\hat{s}_t^\theta$ is summarized as an "advantage" $A$. Looking closely to Equation (16), if $A$ is positive, to increase $J$, the policy making network will be optimized to enhance the probability of action $\hat{a}_t^\theta$ under state $\hat{s}_t^\theta$. In our proposed framework, the actions are within finite selections of the choreographic memory. Hence, the optimization along with Equation (16) is equivalent to the the optimization via an weighted cross-entropy loss on the in-time self predictions of the policy making network:

$$\mathcal{L}_{AC}^{\mathrm{on}} = \sum_{t=0}^{T-1} - \log \left( \pi_\theta(\hat{a}_t^\theta | \hat{s}_t^\theta) \right) \cdot A.$$

(17)

## C.2 ALGORITHM PIPELINE OF PROPOSED OFF-POLICY RL

The detailed pipeline of proposed off-policy RL is shown in Algorithm 1.

## D MORE ABLATION STUDIES

### D.1 MUSIC

To explore the case when the music and the leader are not aligned, we randomize the music input during the test phase. The experimental results are shown in Table 4, where the random music input are noted with ⊠. Interestingly, the BAS of *Duolando* (⊠) even increases by 0.0026 (1.3%) compared to that from the correct musics, while BED score decreases by 0.0276 (10%), implying the music signal have a significant impact on the rhythm of generated follower's motion in our proposed framework.

### D.2 LOOKING AHEAD

We conduct an ablation study on the looking-ahead mechanism, and we train a variant "Duolando w/o. RL LA" where the LAT is substituted to single linear layer to map each token's embedding to the same dimension as LAT (see Table 5). As to the effect of *looking ahead*, compared with "Duolando w/o. RL", the variant without LAT shows a salient drop on $\mathrm{FID}_k$ (26.01, 20%) while the interactive metrics do not significantly change (less than 7%). We observe that lacking the ability to look ahead could lean to some sudden changes in the motion flow, making the movement less smooth.

### D.3 SKATING EFFECT

To assess the severity of the skating effect in generated follower motion, we introduce a new metric, the *skating ratio* (SR), which represents the proportion of frames exhibiting skating. Specifically, if the speeds of the motions of both legs (represented by the vector from the ankle to the pelvis) are less than a threshold $\chi_1$, and the global speed at root (pelvis) is greater than a threshold $\chi_2$, we

Table 5: **Ablation study on *looking ahead* (LA).**

| | Solo Metrics | | | | Interactive Metrics | | | | Rhythmic |
|---|---|---|---|---|---|---|---|---|---|
| Method | $FID_k(\downarrow)$ | $FID_g(\downarrow)$ | $Div_k(\uparrow)$ | $Div_g(\uparrow)$ | $FID_{cd}(\downarrow)$ | $Div_{cd}(\uparrow)$ | CF(%) | BED($\uparrow$) | BAS($\uparrow$) |
| Ground Truth | 6.56 | 6.37 | 11.31 | 7.61 | 3.41 | 12.35 | 74.25 | 0.5308 | 0.1839 |
| *Duolando* w/o. RL LA | 132.73 | 40.22 | 12.94 | 6.70 | 20.80 | 9.92 | 55.81 | 0.2775 | 0.2121 |
| *Duolando* w/o. RL | 106.72 | 34.10 | 13.88 | 7.03 | 21.68 | 9.33 | 57.43 | 0.2795 | 0.2193 |

Table 6: **Skating ratio (SR) of different methods.** The SR is calculated as the percentage of the number of frames with negligible leg movement but significant global displacement.

| Method | SR (% $\downarrow$) |
|---|---|
| Ground Truth | 0.00 |
| *Bailando* (Siyao et al., 2022) | 0.00 |
| EDGE (Tseng et al., 2023) | 0.06 |
| *Duolando* w/o. RL $tr$ IC | 0.00 |
| *Duolando* w/o. RL $tr$ | 0.01 |
| *Duolando* w/o. RL LA | 1.24 |
| *Duolando* w/o. RL | 1.06 |
| ***Duolando*** | 0.33 |

consider this count as a skating frame. It's worthy to note that in calculating global movement, we only consider movement on the $x - y$ plane to account since there exist reasonable vertical shifts (*e.g.*, jumping or lifting) that don't need stepping. Based on the statistics in Figure 7, we set $\chi_1$ to 1 cm per frame and $\chi_2$ to 3 cm per frame. As shown in Table 6, due to the high quality of mocap collection, the ground truth statistically does not exhibit skating. Meanwhile, the SRs of methods like *Bailando*, "*Duolando* w/o. RL $tr$ IC" and "Duolando w/o. RL $tr$" that directly predict displacement based on steps, are also near to zero. In contrast, after introducing $tr$, the frequency of skating steps in "*Duolando* w/o. RL" increases to $1.06\%$. Despite the follower's position being more coordinated with the leader in this variant, the skating effect becomes a significant drawback. To address this issue, we employ RL to align the lower body movement to global shift. As a result, the SR was significantly reduced by $0.73$ ($69\%$), demonstrating the effectiveness of the proposed off-policy RL.

## E    USER STUDY SCREENSHOTS

We conduct our user study through Google Form. The screenshot is shown in Figure 8.

## F    DATA LICENCE

Code and data will be publicly available upon acceptance. The dataset contains the dance sequences in SMPL-X (Pavlakos et al., 2019) format stored in `.npy` files and the corresponding music clips in `.mp3` format. To protect performers' facial information, we do not release the original raster videos following the contracts with performers.

## G    VIDEO DEMO

We provide data examples and generated results in the demo.

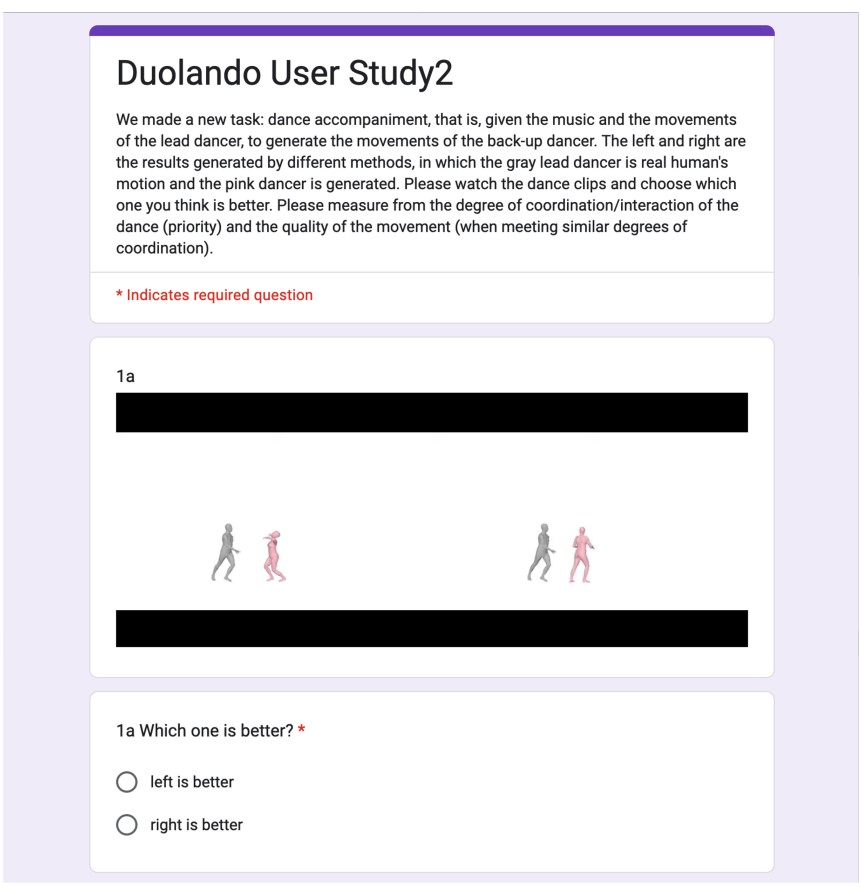

Figure 8: **Screenshot of user study.**

