# OpenReview forum: "Duolando: Follower GPT with Off-Policy Reinforcement Learning for Dance Accompaniment"
_ICLR.cc/2024/Conference — ICLR 2024 poster_

### Official Review · Reviewer_JwGL · 2023-10-30

**Soundness:** 4 excellent
**Presentation:** 3 good
**Contribution:** 4 excellent
**Rating:** 8
**Confidence:** 4

**Summary:**

This work introduces a novel task in 3D dance generation called dance accompaniment. The task involves generating responsive movements from a dance partner, or follower, synchronized with the lead dancer's movements and the underlying musical rhythm. The authors also introduce a large-scale and diverse duet interactive dance dataset called DD100, recorded from professional dancers' performances. A GPT-based model named Duolando is proposed, which predicts motion sequences based on coordinated information from music, leader movements, and previous follower sequences. The model is further enhanced with an off-policy reinforcement learning strategy to generate stable results in unseen conditions.

**Strengths:**

(1)	The paper introduces a new task called dance accompaniment.

(2)	The DD100 dataset containing 115.4 minutes is collected for task training and evaluation.

(3)	The proposed Duolando model utilizes a GPT-based approach to predict follower movements based on coordinated information from music, leader movements, and previous follower sequences.

(4)	The incorporation of an off-policy reinforcement learning strategy enhances the model's ability to generate stable results on unseen conditions.

**Weaknesses:**

This work introduces a new dance accompaniment task, which is interesting and promising for dance generation community. I have some questions and suggestions.

(1)	To gain a better understanding of the differences between various dance styles within the dataset and across datasets, it would be beneficial to provide more data statistics, such as movement speed and action distribution. More details on data processing would be helpful to understand it.

(2)	In Equation 1, it's stated that both p and M are inputs, but in Figure 3, only p is shown as an input.

(3)	It would be interesting to explore which signal, between music and leader motion, has a more dominant influence. For instance, when swapping the music between two test cases in the test set, and using music(B) with leader motion(A) and music(A) with leader motion(B) as inputs, how does this affect the results?

(4)	There is some ambiguity in the terminology between "duet dance generation" and "dance accompaniment." It's important to differentiate them more clearly in the draft. Additionally, it's worth noting that this dataset potentially holds promise for duet dance generation, where only music serves as input to generate movements for two individuals.

**Questions:**

See weaknesses.

(1)	More details on data processing and Figure 3.

(2)	Experiment to explore the importance of signal.

(3)	Questions in writing.

---

> ### Author Response · Authors · 2023-11-20
> **Response to Reviewer JwGL**
>
> > **Q1: "it would be beneficial to provide more data statistics, such as movement speed and action distribution"**
>
> **A1**: We compute the statistics of speed distributions for different types of dances. This part is revised in Appendix B.1 (pp.14).
>
>
> > **Q2: "In Equation 1, it's stated that both p and M are inputs, but in Figure 3, only p is shown as an input"**
>
> **A2**: The VQ-VAE is designed to encode $p$ and subsequently decode it into both $p$ and $M$. $p$ and $M$ in Equation 1 are ground truth targets in the learning, not inputs.
>
> > **Q3: "It would be interesting to explore which signal, between music and leader motion, has a more dominant influence."**
>
> **A3**: As required by the reviewer, we randomize music during the testing phase. The experimental results are shown in Table 4 （pp.17 Appendix D.1). For “Duolando w/o. RL”, incorrect music input causes a reduction in all motion metrics and music metrics. Specifically, the BAS decreases by 12%, while the Beat Echo Degree (BED) decreases relatively less (8%), indicating that leader motion is more crucial in our proposed framework.
>
> > **Q4: "There is some ambiguity in the terminology between 'duet dance generation' and 'dance accompaniment'."**
>
> **A4**: We appreciate the reviewer indicating that. We have revised our paper.

---

> > ### Comment · Reviewer_JwGL · 2023-11-22
> > **Responses to the Authors**
> >
> > Great, my questions are well addressed. Thanks for the authors’ responses.

---

> > > ### Author Response · Authors · 2023-11-23
> > > **Response to Reviewer JwGL**
> > >
> > > We are glad we resolved your concerns. Thank you for your recognition of our work!

---

### Official Review · Reviewer_CR8y · 2023-10-31

**Soundness:** 2 fair
**Presentation:** 2 fair
**Contribution:** 1 poor
**Rating:** 3
**Confidence:** 4

**Summary:**

The paper presents a method for generating dance accompaniment with respect to a leader dancer dancing under a music piece. The method consists of a series of VQVAEs for encoding body motions, music encoder, and a GPT for autoregressive modeling the conditions and follower tokens. In addition, an off-policy RL method is employed to improve the generation quality. In terms of the dataset, the authors collect a duet dance dataset from professional dancers along with the music.

**Strengths:**

1. The paper presents a novel task, generation of a dance follower conditioned on the other dance along with the music.

2. The author contributes a dataset and the motion part is accurately captured using MoCap sensor.

**Weaknesses:**

1. How is music represented? Midi or spectrogram or raw waveform? I didn’t see where the paper defined the representation of it.

2. An important ablation study is missing, what if the music is not fed as input. I think it still makes sense to generate a follower dancer as the rhythm of music is already embedded in the leader dancer?

3. The RL setup aims to handle the OOD data for unseen dance motions, and the authors present an off-policy learning approach. However, I didn’t see ablations against on-policy RL nor ablations on the reward design, which looks very ad-hoc to me.
The supplementary material provides videos for original dataset and generated samples. However, I find some samples where actions are not well-aligned with music beats at all, I doubt whether it is something performed by professional dancers. Also, from the generated samples, I saw that sometimes part of a body model passing through part of another body model in an unnatural manner.

**Questions:**

See weakness.

---

> ### Author Response · Authors · 2023-11-20
> **Response to Reviewer CR8y**
>
> > **Q1: "How is music represented? "**
>
> **A1**: We apologize for the oversight.
>
> *The music features $\bf{\it m}$ are extracted from the audio signal using the public audio processing toolbox Librosa (McFee et al. 2015), including mel frequency cepstral coefficients (MFCC), MFCC delta, constant-Q chromagram, onset strength and onset beat, which are 54-dimension in total, and are mapped to the same dimension （438）of GPT via a learned linear layer.*
>
>
> This part was originally in the main text but was inadvertently omitted when moving it to the appendix due to page limitation. We have added this back in the revised version (pp.15 Appendix B.2).
>
> > **Q2: "What if the music is not fed as input?"**
>
> **A2**: (1) To explore the effect of the music signal input, we train a new variant GPT without the slots for music: ***Duolando w/o. RL*** $m$. The experimental results have been updated in the revised draft (Table 4, Appendix D.1, pp. 18). Without the music input, the follower achieves a comparable Beat-Align Score (BAS) to that with music input since the follower is explicitly anchored to the leader's position and will follow the rhythm indirectly.
>
> (2) The above alignment is under the condition that the follower's motion aligns with the music well. In the other experiment, we randomized the music input in the test phase. As shown in Table 4, the BAS of ***Duolando w/o. RL*** $m$ is 0.1657, lower than that with music input (0.1840), which shows the music encoding will significantly impact the motion when music is not aligned with the movement.
>
>
>
> > **Q3: "... ablations against on-policy RL nor ablations on the reward design"**
>
> **A3**: (1) We present the reward curves of the proposed off-policy RL and the on-policy AC (Siyao et *al.*, 2022) on test set in Figure 8, pp.18 and Appendix D.2, pp. 19. The average sequantial reward for the off-policy RL shows a steady increase; in contrast, the on-policy RL trend is less stable and even collapses at around the 30-th epoch. This aligns with the phenomena reported in (Siyao et *al.*, 2022) that excessive fine-tuning in on-policy AC leads to instability.
>
> (2) Given that our reward system comprises only a single reward, conducting a compositional ablation study is impractical. On the reward itself,  we introduced a threshold of for the penalty. If without this threshold, the follower tend to perform excessively redundant small steps to align the translation strictly.
>
>
>
> > **Q4: "... whether ... performed by professional dancers"**
>
> **A4**: All dancers involved in our recordings are professionally trained and have won awards in professional competitions. After a thorough comparison, we confirm that the segments provided in our demo align with the live video footage. The perception of misalignment between motion and music that the reviewer notes might be attributed to the concept of 'rubato'. In ballroom dancing, particularly in smooth styles, dancers may intentionally delay a step or movement to coincide with a specific musical element or to create a desired effect or expression. This artistic choice enhances the performance's expressiveness but might appear as a misalignment to those unfamiliar with the technique.
>
> > **Q5: "... outputs have penetration."**
>
> **A5:** Indeed. As the first solution to dance accompaniment, while our method is not yet perfect, it effectively tackled some key issues through the incorporation of Interaction Coordination (IC), relative translation (tr), and Reinforcement Learning (RL). These improvements resolve issues like action irrelevance, inappropriate relative displacement, and follower's lower body incompatibility, making our approach significantly superior to existing solo dance methods in the interactive scenarios. It can serve as a solid baseline for this newly proposed task.

---

> ### Author Response · Authors · 2023-11-23
>
> Dear Reviewer CR8y,
>
> We sincerely thank your great effort in reviewing our work. We have improved and revised our submission based on your comments. As the deadline is approaching, please let use know if you have any follow-up concerns. We sincerely hope you can consider our reply in your assessment. Thank you!
>
> Sincerely, ICLR 2024 Conference Submission2432 Authors

---

> > ### Comment · Reviewer_CR8y · 2023-11-23
> >
> > I thank the authors for additional clarifications on music reprsentation, ablation study of music tokens. However, as an application paper, the quality of generated dance movements is still far from satisfactory and also the scale and quality (after carefully inspection of the samples again) of the collected dataset is still a concern. I choose to maintain my rating.

---

> > > ### Author Response · Authors · 2023-11-23
> > > **Response to Reviewer CR8y**
> > >
> > > As the first solution to dance accompaniment, our method tackles some key issues like action irrelevance, inappropriate relative displacement, and follower's lower body incompatibility, making our approach significantly superior to existing solo dance methods in the interactive scenarios.
> > >
> > > As to the dataset, the proposed DD100 contains over 1.9 hours high quality mocap data of professional duet dancers, including body and hand (fingure-level) motions. The dataset includes strong interactions in long duration (> 1 min in average), which are not in existing works.
> > >
> > > We sincerely wish the reviewer consider these into assessment.

---

### Official Review · Reviewer_otBs · 2023-10-31

**Soundness:** 3 good
**Presentation:** 3 good
**Contribution:** 3 good
**Rating:** 8
**Confidence:** 4

**Summary:**

- This paper proposed an interesting human-human interaction task called dance accompaniment generation, which aims to generate human dancing motions conditional on a leader dance and background music.
- Identifying the need for a dataset for this new task, the author proposed a mo-caped duet dancing dataset, containing 1.92 hours of duet dancing, with an average time for each clip over 1 minute. The dataset is collected with 20 120-FPS optical cameras and meta gloves for motion capture, and it contains 10 distinct genres of duet dances. SMPL-X is fitted for this dataset.
- This paper proposed a pure kinematics-based method for conditional dance generation. It first tokenizes the dancing motion with 5 VQ-VAE, then trains a GPT-like arch to generate dancing motions autoregressive. To reduce the misalignment of lower-body motion and global root translation, the author used off-policy RL to finetune the model.
- For experimental evaluations, the author proposed a set of simple metrics for evaluating the follower's interaction with the leader and the background music. The metric is inspired by the Beat-Align Score, where the beat time of dance is defined as the local minimum of velocity. Besides metrics, a user study is also conducted. The author showed better performance of its proposed algorithm than other adopted baselines.

**Strengths:**

- The algorithm proposed does not change the whole landscape of the kinematics-based motion generation, but it has some novelty, especially the second-stage off-policy tuning. Additionally, the authors also collected a high-quality duet dancing dataset, making this paper more sound.
- For the evaluation part, it is thorough to design three separate metrics and also a user-study.
- The writing, tables, and figures are clear and easy to follow. For example, table-1 cleary summerize the unique of the collected dataset. Figure-4 shows the autoregressive generator quite clearly. The details of the methods are well-documented in the supplementary parts.

**Weaknesses:**

I only have some minor comments on the weakness part.

- I found a major weakness in the off-policy RL part. For a pure kinematics-based method, there might exist a couple of artifacts, such as floor-skating, penetration between bodies and ground, and other physical-infeasible dynamics. The videos in the supplementary materials also showed such artifacts. The paper only addresses the problem of misalignment of lower-body motion and full-body motion; why the author just stopped here? it seems that other artifacts might also be alleviated through similar RL tuning. Has the author tried to address such artifacts? Does the author believe such off-policy tuning combined with some heuristic reward function can reduce most artifacts?
- The author mentioned using off-policy RL to reduce the artifacts of misalignment of lower and full-body motion. It would be beneficial to add some new metrics to evaluate this specific improvement.

**Questions:**

- Why use 2D skeletons for user study? Why not use 3D-rendered motions, like the one in the demo video?
- What is the training time for each of the stages?
- A general question is how to evaluate the quality of the motion-capture? I watched the video, and it seems there are some flickering frames and some exaggerated hand motions.

---

> ### Author Response · Authors · 2023-11-20
> **Response to Reviewer otBs**
>
> > **Q1: "The paper only addresses the problem of misalignment of lower-body motion and full-body motion; why the author just stopped here? ... Has the author tried to address such artifacts? Does the author believe such off-policy tuning combined with some heuristic reward function can reduce most artifacts?"**
>
> **A1**: We appreciate the reviewer's suggestions for a more in-depth exploration of the RL rewards to enhance dance quality.
>
> (1) Our primary focus in this work is to enable the follower to dance responsively to the leader. Beginning with a solo dance GPT structure similar to *Bailando* (Siyao et *al.*, 2022), we employ Interaction Coordination (IC) and explicit relative translation (***tr***) to make the follower correspond with the leader on movement and positioning, respectively. The proposed RL concentrates on solving/suppressing the side effect of ***tr*** -- the skating effect, caused by misalignment between lower body and global shift.
>
>
> Regarding penetration issues, potential post-processing exists in the industry and the academy [1,2,3]. A feasible RL solution can be rewarding non-intersection while penalizing the corresponding intersecting parts based on penetration depth. For instance, intersections on hands and arms should lead to a penalty on the upper body, while that on the trunk should impact translation (***tr***).
>
> (2)  We acknowledge that RL alone may not resolve all issues. In practical applications, graphical post-processing is expected for further quality enhancement.
>
> #### Reference
>
>
> [1] Smith et *al.* Open dynamics engine, 2005
>
> [2] Pavlakos et *al.*. Expressive body capture: 3d hands, face, and body from a single image. CVPR 2019.
>
> [3] Müller et *al.* On self-contact and human pose. CVPR 2021.
>
> > **Q2: "It would be beneficial to add some new metrics to evaluate this specific improvement (of RL)."**
>
> **A2**: To show effect of RL on reducing skating effect, we propose a metric named *slipping ratio* (SR). Specifically, we compute the percentation of the length where the feet/legs do not move obviously but the global shift is significant (> 1 cm/frame). The results are shown in Table 4, Appendix D.3, pp.19. Experimental results show our proposed RL can reduce over 60% of skating in that without RL.
>
> > **Q3: "Why not use 3D-rendered motions (for user study)?"**
>
> **A3**: As required by the reviewer, we conduct a new user study using 3D-rendered SMPL-X models. We also updated the results on the revised draft. Generally, the average winning rates do not change greatly and do not change the conclusion.
>
> > **Q4: "What is the training time for each of the stages?"**
>
> **A4**: The training time for VQ-VAEs is about 2 days. The training time of GPT in the supervised learning stage is around 1.5 days and that of the RL stage is about 18 hours. Four NVIDIA V100 GPUs are used for training.
>
> > **Q5: "How to evaluate the quality of the motion-capture? It seems there are some flickering frames and some exaggerated hand motions"**
>
> **A5**: Thanks for bringing up the issues with the mocap evaluation. We have observed similar phenomena in our data and have undertaken methods to refine mocap data quality along two dimensions:
>
> 1. **Flickering frames**: Inaccuracies arise when marker points are potentially obscured during the collection process, resulting in inaccuracies in specific frames. To address this, we introduce two kinds of information: (1) **Single-frame confidence**. We leverage marker point confidence during the fitting process, excluding points with low confidence. (2) **Temporal smoothness.** If most of the markers within a frame have low confidence, the fitting outcome may be suboptimal. We introduce temporal information in post-processing to first detect abrupt changes from the temporal axis. Subsequently, we eliminate such frames and employ interpolation to complete the missing frames.
> 2. **Exaggerated hand motion**: This issue typically arises from either accumulated errors in the original glove data or a lack of constraint information for joints. To avoid this, explicit constraints are imposed on each finger and wrist joint according to human anatomy. For instance, finger joints are restricted from undergoing self-rotation, and hand wrists are also constrained against self-rotation.).

---

> ### Author Response · Authors · 2023-11-23
>
> Dear Reviewer otBs,
>
> We sincerely thank your great effort in reviewing our work. We have improved and revised our submission based on your comments. As the deadline is approaching, please let use know if you have any follow-up concerns. We sincerely hope you can consider our reply in your assessment. Thank you!
>
> Sincerely, ICLR 2024 Conference Submission2432 Authors

---

### Official Review · Reviewer_cNc4 · 2023-10-31

**Soundness:** 4 excellent
**Presentation:** 4 excellent
**Contribution:** 2 fair
**Rating:** 6
**Confidence:** 3

**Summary:**

This paper proposes a new task and motion captured dataset related to dance generation, specifically, the task of generating dance “accompaniments”. In this task, the model’s job is to generate the follower’s motion from that of the leader, potentially enabling VR/AR experiences where users dance with virtual avatars. This paper also proposes a GPT-based method to address this task which treats dance accompaniment as a sequence-to-sequence language modeling task to convert leader dance tokens and musical context into follower dance tokens.

**Strengths:**

This paper has several strengths including **introducing a novel task and dataset**, **proposing well-designed methods with solid evaluation**, and **it is well-written**.

**Novel task and dataset**. The idea of generating dance accompaniments is interesting, and the authors expend extraordinary effort and expense to create a novel dataset for this task. Based on the supplementary material, this dataset appears to my eyes to be of extremely high quality thanks to the use of fine-grained motion capture. I have no doubt that this dataset will constitute a valuable resource to the growing dance generation research community.

**Well-designed methods and evaluation**. This paper poses dance accompaniment as a sequence-to-sequence “text generation” problem by learned tokenizations of dance motion. Despite the inevitable methodological complexity that comes with dealing with high-dimensional dance data, this approach is _overall_ satisfyingly straightforward and reasonable. Moreover, the authors construct reasonable quantitative evaluation metrics for their method and also conduct a user study, achieving impressive performance relative to baselines (but less impressive than the ground truth) and also effectively ablating additional elements of their approach (e.g. RL).

**Well-written**. This paper is also very well-written, with remarkable clarity both in its conceptual presentation and formalized notation. I was able to follow the details quite well despite being a newcomer to working with dance and motion capture data.

**Weaknesses:**

This paper also has some weaknesses including **output quality issues**, **potential copyright issues with the dataset**, and **limited reusability of insights**.

**Output quality issues**. While reasonable in design, the proposed method produces fairly rigid dance accompaniments that, subjectively speaking, vaguely resemble someone dancing with a lifeless mannequin (especially in contrast to extraordinary richness of the ground truth accompaniments). While the authors make a valiant effort to improve results w/ RL, the best system still falls far short. It seems like progress here is more likely to be driven by large-scale pre-training from noisier dance datasets followed by adaptation to small high-quality mocap datasets, rather than training from scratch on mocap datasets.

**Potential copyright issues with the dataset**. Perhaps moreso than the proposed methods, the DD100 dataset may be the most valuable aspect of this work to the dance generation community. This paper promises to release “MP3s” associated with the dataset but fails to report details about the copyright status of the music in the dataset. Listening to the demo videos, the dataset appears to feature copyrighted material (e.g. “Charlie Puth - LA Girls”). It is likely that, even if the authors release the audio, they will likely be forced to take it down eventually, compromising the value of the dataset to the research community.

**Limited reusability of insights**. There is not a lot of information in this paper that would be of interest to researchers outside of the dance generation community. Perhaps there is something reusable happening in the use of RL to refine GPT models, but this is only explored within the context of dance generation. Though ICLR does occasionally have dance generation work in its proceedings, I suspect that this paper overall will not be particularly interesting to the broader ICLR community.

**Questions:**

- Why use copyrighted music for capturing this valuable dataset as opposed to copyright-free audio?
- Why not pre-train models on noisy dance data?
- There doesn’t appear to be any details about the music tokenization strategy used in this work - can the authors clarify how music features are represented?
- I was confused by the use of lookahead - would the lookahead model actually work in a real-time dance setting?

**Details Of Ethics Concerns:**

Dataset appears to use copyrighted MP3s. From an _ethical_ point of view, my personal stance is that this is fine: the goal of this dataset is clearly to provide realistic dance data set to realistic music moreso than a subversive effort to distribute a few dozen copyrighted songs. However, from a _legal_ point of view, it is likely that the music in the dataset will have to be taken offline at some point, harming its usefulness as a research tool.

Another mild ethical concern revolves around the gendered nature of the application goal of generating *follower* dancers, conventionally the role of female dance partners. The authors could have also studied generating leader motions, a similarly interesting task that would allow users to dance with leaders. I would like to see the authors briefly address this in their ethical statement as well.

---

> ### Author Response · Authors · 2023-11-20
> **Response to Reviewer cNc4**
>
> > **Q1: “Output quality issues”**
>
> **A1**: As the first solution to dance accompaniment, while our method is not yet on par with professional human dancers, it effectively tackled some key issues through the incorporation of Interaction Coordination (IC), relative translation (tr), and Reinforcement Learning (RL). These improvements resolve issues like action irrelevance, inappropriate relative displacement, and follower's lower body incompatibility, making our approach significantly superior to existing solo dance methods in the interactive scenarios. It can serve as a solid baseline for this newly proposed task.
>
> > **Q2: “Potential copyright issue of the music mp3… why use copyrighted music”**
>
> **A2**: **Choice of Music.** These tracks were chosen by the performers from their familiar collections to ensure the quality of performance.
>
> **‘Fair Use’ [1, 2] Justification.** The music segments are modified (clipped, and partially slowed down), deviating from their original form. These modifacations, coupled with their use solely for model training/testing rather than entertainment, comply with fair use criteria. Additionally, the academic nature of our work and the absence of public searchability (we won’t provide music info beyond the audio) avoid potential commercial impact. Therefore, the music included in our dataset falls under ‘fair use’ provisions, while it also align with previous practices [3, 4] in dance geration domain, presenting a relatively low risk. We have also updated these facts in the ethical statement in our paper.
>
> **Contingency Plans.** Should the need arise to remove certain tracks, we can provide them in a non-playable, digitized format suitable for model training/testing. Or at least, we can offer only the titles and relevant details (like YouTube links and specific segment/speed information) with automatically downloading/processing scripts, a common method used in prior academic datasets [5, 6].
>
> #### Reference
>
> [1]https://guides.nyu.edu/fairuse#:~:text=Fair%20use%20allows%20limited%20use,automatically%20qualifies%20as%20fair%20use
>
> [2] https://en.wikipedia.org/wiki/Fair_use
>
> [3] Lee et *al.*, Dancing to Music, NeurIPS2019
>
> [4] Huang and Hu et *al.*, Dance Revolution: Long-Term Dance Generation with Music via Curriculum Learning, ICLR 2021
>
> [5] Siarohin et *al.*, First Order Motion Model for Image Animation, NeurIPS 2019
>
> [6] Miao et *al.*, VSPW: A Large-scale Dataset for Video Scene Parsing in the Wild, CVPR 2021
>
> > **Q3: “Limited reusability of insights (to broader community other than dance generation).”**
>
> **A3**: Our method can extend beyond dance generation and offer insights for human-human interaction tasks, a key area of 3D human motion and virtual reality. Additionally, it holds promise for integrating 3D human interaction within unified language models (GPT), which could interest a broader range of researchers.
>
> Furthermore, our proposed DD100 dataset contributes strong interaction data with long average durations to the human-human interaction research community.
>
> > **Q4: “Why not pre-train models on noisy dance data”**
>
> **A4**: This is indeed a great idea. However, at present, there is a lack of such large-scale noisy data on interactive duet dance (e.g., that from monocular video reconstructions).
>
> > **Q5: “There doesn’t appear to be any details about the music tokenization strategy used in this work - can the authors clarify how music features are represented?”**
>
> **A5**: We apologize for the oversight.
>
> *The music features $\bf{\it m}$ are extracted from the audio signal using the public audio processing toolbox Librosa (McFee et al. 2015), including mel frequency cepstral coefficients (MFCC), MFCC delta, constant-Q chromagram, onset strength and onset beat, which are 54-dimension in total, and are mapped to the same dimension of GPT via a learned linear layer.*
>
> This part was originally in the main text but was inadvertently omitted when moving it to the appendix due to the page limitation. We have added this back to the revised version (pp.15 Appendix B.2)
>
> > **Q6: “I was confused by the use of lookahead - would the lookahead model actually work in a real-time dance setting”**
>
> **A6**: The 'look-ahead' feature views future music and leader actions (4 seconds ahead in our work) to generate smoother movements, as demonstrated in Table 2 (Duolando w/o. RL LA) and in the supplementary video at 7'39". However, it indeed does not yet meet real-time requirements.
>
> A potential solution could be to predict future music and leader motions in advance rather than viewing them from the data.

---

> > ### Author Response · Authors · 2023-11-23
> >
> > Dear Reviewer cNc4
> >
> > We sincerely thank your great effort in reviewing our work. We have improved and revised our submission based on your comments. As the deadline is approaching, please let use know if you have any follow-up concerns. We sincerely hope you can consider our reply in your assessment. Thank you!
> >
> > Sincerely,
> > ICLR 2024 Conference Submission2432 Authors

---

> > > ### Comment · Reviewer_cNc4 · 2023-12-05
> > >
> > > Apologies for my inexcusably delayed response. The authors addressed many of my concerns in their rebuttal and paper updates. I remain concerned that the results of proposed method are subjectively quite poor, though I agree with the authors that it represents a reasonable baseline. I am not enough of a legal expert to evaluate the authors' fair use justification (nor is jurisdiction clear), but I certainly sympathize with the argument from an ethical point of view. Given all of this, I will raise my score from weak reject to weak accept.

---

### Author Response · Authors · 2023-11-20
**Response to all reviewers**

Dear reviewers,

We would like to sincerely thank your comments on our submission. We have responded to your concerns point-by-point in the comments section below. The revised parts in the paper are marked in cyan color.

If you have further comments, please follow up below. If you feel our responses have addressed your concerns, we hope you may consider increasing your score.

Thank you!

Sincerely,
ICLR 2024 Conference Submission2432 Authors

---

### Meta-Review · Area_Chair_j7sV · 2023-12-07

**Metareview:**

The paper addresses a novel task, 3D dance accompaniment, which requires neural dancer to dance synchronically with the lead dancer's movement as well as the music. The paper receives three strong positive reviews and a negative reviews. The positive reviews are given as the proposed method, despite simple, is well-designed and evaluated. The negative reviews majorly lie in (1) whether the scope is of the interest of ICLR (2) some missing ablation and analysis. (2) has been addressed in the rebuttal phase. As for the scope of ICLR, I believe that despite the paper is application-driven, the problem formulation and the solution are valuable to the community.

**Justification For Why Not Higher Score:**

A big part of the contributions lie in the problem setup, the proposed approach is rather simple.

**Justification For Why Not Lower Score:**

The task is novel and the proposed method is effective. Thorough experiments are provided.

---

### Decision · Program_Chairs · 2024-01-16

Accept (poster)